# The Spectrum of CAR Cellular Effectors: Modes of Action in Anti-Tumor Immunity

**DOI:** 10.3390/cancers16142608

**Published:** 2024-07-22

**Authors:** Ngoc Thien Thu Nguyen, Rasmus Müller, Daria Briukhovetska, Justus Weber, Judith Feucht, Annette Künkele, Michael Hudecek, Sebastian Kobold

**Affiliations:** 1Division of Clinical Pharmacology, Department of Medicine IV, LMU University Hospital, LMU Munich, 80336 Munich, Germany; thu.nguyen@med.uni-muenchen.de (N.T.T.N.); rasmus.mueller@med.uni-muenchen.de (R.M.); daria.briukhovetska@med.uni-muenchen.de (D.B.); 2German Cancer Consortium (DKTK), Partner Site Munich, a Partnership between the DKFZ Heidelberg and the University Hospital of the LMU, 80336 Munich, Germany; 3Department of Medicine II, Chair in Cellular Immunotherapy, University Hospital Würzburg, 97080 Würzburg, Germany; weber_j3@ukw.de (J.W.); hudecek_m@ukw.de (M.H.); 4Cluster of Excellence iFIT “Image-Guided and Functionally Instructed Tumor Therapies”, University of Tübingen, 72076 Tuebingen, Germany; judith.feucht@med.uni-tuebingen.de; 5Department of Hematology and Oncology, University Children’s Hospital Tuebingen, University of Tübingen, 72076 Tuebingen, Germany; 6Department of Pediatric Oncology/Hematology, Charité-Universitätsmedizin Berlin, Corporate Member of Freie Universität Berlin and Humboldt-Universität zu Berlin, 13353 Berlin, Germany; annette.kuenkele@charite.de; 7German Cancer Consortium (DKTK), Partner Site Berlin, 10117 Berlin, Germany; 8Fraunhofer Institute for Cell Therapy and Immunology, Cellular Immunotherapy Branch Site Würzburg, 97080 Würzburg, Germany; 9Einheit für Klinische Pharmakologie (EKLiP), Helmholtz Zentrum München—German Research Center for Environmental Health Neuherberg, 85764 Oberschleißheim, Germany

**Keywords:** adoptive cell therapy, chimeric antigen receptor, mode of action, natural killer cells, macrophages, gamma-delta T cells, dendritic cells, kinetics, persistence, killing mechanism, cytokines, chemokines

## Abstract

**Simple Summary:**

Chimeric antigen receptor (CAR)-T cells have revolutionized the treatment of blood cancers. However, their effectiveness meets challenges in solid tumors. Modifications to CAR-T cells altered stimulatory domains or added further functions with transgenic proteins (e.g., cytokines, chemokine receptors, degrading enzymes) to match specific barriers of the tumor microenvironment. But despite these enhancements, the inherent limitations of CAR-T cells such as dependency on human leukocyte antigen (HLA), toxicities and high costs for preparing autologous cell products remain. In response, alternative types of immune cells with different effects and advantages have become the focus of adoptive cell therapy research. For instance, natural killer cells have the benefit of HLA-independent killing and macrophages possess additional functions such as phagocytosis and antigen presentation. As these cells come with distinct properties, clinicians and researchers need a thorough understanding of their effects and peculiarities. This review summarizes the different modes of action of these CAR-immune cells.

**Abstract:**

Chimeric antigen receptor-T cells have spearheaded the field of adoptive cell therapy and have shown remarkable results in treating hematological neoplasia. Because of the different biology of solid tumors compared to hematological tumors, response rates of CAR-T cells could not be transferred to solid entities yet. CAR engineering has added co-stimulatory domains, transgenic cytokines and switch receptors to improve performance and persistence in a hostile tumor microenvironment, but because of the inherent cell type limitations of CAR-T cells, including HLA incompatibility, toxicities (cytokine release syndrome, neurotoxicity) and high costs due to the logistically challenging preparation process for autologous cells, the use of alternative immune cells is gaining traction. NK cells and γδ T cells that do not need HLA compatibility or macrophages and dendritic cells with additional properties such as phagocytosis or antigen presentation are increasingly seen as cellular vehicles with potential for application. As these cells possess distinct properties, clinicians and researchers need a thorough understanding of their peculiarities and commonalities. This review will compare these different cell types and their specific modes of action seen upon CAR activation.

## 1. Introduction

Advances in biomolecular and immunological research in recent decades have dramatically changed cancer therapy. The understanding that each cancer cell type is unique, with different antigen expressions, specific genetic mutations and varying pathway abnormalities, has led to a more individualized tumor treatment, and the three original pillars of cancer therapy—surgery, chemotherapy and radiotherapy—have been reinforced by targeted and immunotherapy.

Immunotherapy uses the immune system to treat cancer. It includes immune system-modulating agents such as immune checkpoint inhibitors, vaccines and cytokine infusions, or direct immune engaging agents such as tumor-targeting antibodies and adoptive cell transfer.

Modern cancer immunotherapy is based on the findings in the 1950–1960s that lymph node cells were able to induce the neutralization of sarcoma cells in vitro and cause the rejection of fibrosarcoma in mice [1,2]. These findings are the founding stones of cellular immunotherapy.

T-lymphocytes (T cells) were soon after identified as the main effectors of antigen-dependent immunologic anti-tumor response [3]. The successful isolation of tumor-infiltrating lymphocytes (TILs) followed, and in 1986, ex vivo expanded TILs were injected as a living drug into mice bearing MC-38 colon adenocarcinoma cells, which in combination with interleukin-2 (IL-2) and cyclophosphamide or total body irradiation cured the mice of metastatic lesions in the liver and lung [4]. This study paved the way for adoptive cell therapy in cancer treatment, resulting in the first clinical trial on adoptive cell therapy in humans in 1988 [5]. TILs, however, were hard to isolate and cultivate, and could only be used to treat the tumor they were isolated from; they were not transferable to other malignancies [3]. With the increase in knowledge about tumor-associated antigens (TAAs), the specific antigen recognition of T cell receptors (TCR) and their structures and gene transfer methods in the following years, researchers strived to induce specificity in naïve T cells, which were abundant in peripheral blood, and easy to isolate and to cultivate [3]. The TCR from a melanoma TIL was therefore cloned and transduced into naïve T cells, and produced successful results in a clinical trial with melanoma patients [6]. This success was, however, restricted to patients with HLA-A*0201 expression, as TCR recognition is HLA-dependent [6]. This was, therefore, the next hurdle to overcome, and resulted in a fusion protein consisting of an antigen recognition domain derived from antibodies and the intracellular CD3ζ signaling domain of the TCR/CD3 complex, creating the first-generation chimeric antigen receptor (CAR) and CAR-T cell, which was then in 1993 called the “T-body” by the Eshhar Group [7,8].

Since then, CARs have been defined as recombinant receptor proteins engineered to be expressed on the immune cell surface in order to redirect the immune response and to target specific cells. The receptor consists of three parts: the extracellular domain contains an antigen recognition domain to target different surface antigens (e.g., single chain variable fragment (scFv)); a hinge and transmembrane region gives the CAR the range of motion to bind to epitopes and anchors it in the membrane; the intracellular signaling domain mostly consists of the CD3ζ domain of the TCR/CD3 complex with the immunoreceptor tyrosine-based activation motif (ITAM) as signal transducer, and can be reinforced by co-stimulatory domains, e.g., CD28 and 4-1BB (CD137), which are mostly used in clinical adoptive cell therapy [9,10]. All these components can be modified to give the CAR different targets with specific binding capacities and access to distal or proximal membrane epitopes, alter signal transduction and duration, or enable the phenotype selection and induction of transgenic protein. All these modification possibilities have produced so-called CAR generations and CAR construct combinations with variable success rates.

The first clinical trial of the anti-CD19-CD137-CD3ζ CAR-T cells in a CLL patient resulted in complete remission in 2010, marking the success of a novel treatment [11]. Further developments led to the FDA approval of the second-generation anti-CD19-CD137-CD3ζ CAR tisagenlecleucel (Kymriah) in 2017 and anti-CD19-CD28-CD3ζ CAR axicabtagene ciloleucel (Yescarta) in 2018. Both CARs showed complete remission rates (CRR) of 40% (JULIET trial) and 58% (ZUMA-1 trial), and overall survival (OS) of 36% (3-year-OS, JULIET trial) and 42% (5-year OS, ZUMA-1 trial), respectively [12,13,14]. Since then, four further CAR-T cell products have achieved FDA approval. CAR-T cell therapy has developed into an efficacious treatment inducing remission in various hematological malignancies, such as acute lymphoblastic leukemia (ALL), lymphomas and multiple myelomas [12]. This generated a paradigm shift in cancer treatment, and CAR-T cells were included in the guidelines as an inherent part of hematological malignancy treatment.

This success, however, was not transferable to solid tumors, as a range of properties posed significant challenges for CAR-T cells. The growth of solid tumors in cell clusters, instead of isolated cells like in most hematological malignancies, and deficient vascularization impedes CAR-T cell access and trafficking. Heterogeneous antigen expression makes specific targeting a challenge [15]. The unique tumor microenvironment creates a physical barrier with a dense extracellular matrix that is impenetrable for CAR-T cells, whereas an immune-suppressive environment dampens anti-tumor response [16]. Additionally, the metabolically hostile environment increases T cell exhaustion and reduces their persistence [15,16]. Some of these obstacles can be addressed by CAR engineering by increasing activity and persistence, reprogramming exhausted T cells, creating switch receptors or armouring CAR-T cells with transgenic proteins such as cytokines, chemokine receptors or matrix-degrading enzymes [17]. Furthermore, some challenges arise from the cells’ inherent properties, including HLA-incompatibility, decreased tissue affinity and infiltration, limited effector function, toxicity (cytotoxic release syndrome, immune effector cell-associated neurotoxicity syndrome) and the complex preparation process required for autologous cells.

Cell type-associated limitations have shifted other immune cells into the focus of cellular therapy. Cells of the innate immune system, such as macrophages and dendritic cells (DC), are tissue-resident and possess mechanisms to infiltrate and move within the tissue, as well as different effector functions like phagocytosis, antigen presentation and immune system modulation. Natural killer (NK) cells and γδ T cells do not have to be HLA compatible, enabling the use of cell lines and alternative cell sources. But all these different cell types also come with individual properties, making the understanding of their peculiarities and commonalities crucial for their application. This review will discuss and compare the four different cell types, NK cells, macrophages, γδ T cells and DC, as well as their specific modes of action (CAR-activation, kinetics, soluble effector molecules, membrane-bound effector molecules) upon the activation of the CAR in these cell types.

## 2. T-Lymphocytes

There are detailed reviews concerning CAR-T cells’ efficacy, engineering and mode of action. This article will only give a summary of the different modes of action of a CAR-T cell as a comparison reference for the following CAR cell types, and will mainly refer to the review of Benmebarek et al. [18]. Beyond this, detailed information concerning the various killing mechanisms will be provided in the sections below.

T cells are equipped with a CAR to redirect and enhance their cytotoxic abilities. Several CAR generations have evolved, each enhancing another aspect of the CAR-T cell. Currently, at least five generations of CAR can be defined. First-generation CAR fuses scFv directly to an intracellular signaling domain, mostly CD3ζ. Second-generation CARs are additionally equipped with a co-stimulatory domain to ensure persistence and improved effector functions. Third-generation CARs are characterized by multiple co-stimulatory domains. Fourth-generation CARs are equipped with additional genes that encode transgenic proteins such as cytokines, enzymes, or ligands, and the fifth-generation CAR is not yet clearly defined, but encompasses CARs that split the signaling domain and the antigen recognition unit, making the signaling domain stationary and the antigen recognition unit exchangeable to make the CAR more universal and admissible for several antigens, e.g., BBIR (biotin-binding immune receptor) CAR or SUPRA (split, universal, programmable) [19]. However, the fifth generation also contains CARs that express additional receptor domains such as IL-2 receptor domains, which allow JAK/STAT pathway activation [19,20].

CAR activation leads to ITAM phosphorylation in the CD3ζ intracellular domain of the CAR and the recruitment of ZAP70, which induces several pathways, such as actin polymerization via the Vav-Rac pathway; this is crucial for immune synapse formation and degranulation, or cytokine and chemokine release via (phospholipase C gamma) PLCγ, which forms inositol-1,4,5-trisphosphate (IP3) for calcium influx and NFAT translocation into the nucleus, and diacylglycerol (DAG), which promotes NF-κB translocation and together with the Sos-Ras pathway also induces AP-1 formation [21,22,23].

CAR-T cell kinetics consist of a distribution, expansion, contraction and persistence phase. Each phase is highly variable in peak measurements and duration depending on interpatient factors and product-specific factors [24,25]. Biodistribution occurs within days, followed by an expansion phase ending with peak expansion, which can take place between the first 1 and 2 weeks and leads into the contraction (3–6 months) and persistence phases, which can vary up until 5 years [24]. CAR-T cells are mainly distributed to blood and tumors, but they are also found in lungs, liver, spleen, cerebral spinal fluid and bone marrow [24]. Concerning kinetics, some studies show that a low peak expansion correlates with increased therapeutic failure [25], whereas the persistence and long-term persistence of CAR-T cells correlate with durable remission [25]. But there are also studies that have been unable to provide these correlations [25]. An analysis of 92 patients who received axicabtagene ciloleucel (Yescarta), tisagenlecleucel (Kymriah) and brexucabtagene autoleucel (Tecartus) between 2019 and 2022 have been screened for kinetics [25]. It was shown that a median peak expansion was reached within 9 days and did not correlate with tumor response, whereas the cell number peak correlated with tumor response and persistence [25]. Kymriah also showed a higher persistence percentage after 6 months than Yescarta and Tecartus (96% vs. 65%) [25]. In 16% of patients, CAR-T cells were no longer detectable after 6 months, with a median time to undetectable CAR-T cells of 98 days (range: 17–651) [25]. For all products, CAR-T cells were detectable in some patients up until 36 months [25]. Persistence also seemed to be longer in CD137-CD3ζ constructs than CD28-CD3ζ CAR [25]. Altogether, persistence is an indicator tightly linked to clinical efficacy, and can be influenced by T cell phenotype and T cell fitness. T cell subsets such as central memory T cells or stem cell memory T cells were more prevalent in patients with sustained anti-tumor response [26]. The selection of these specific subsets and production of CAR-T cells with defined CD4+ and CD8+ phenotypes and compositions can therefore provide uniform potency and superior antitumor activity [27,28,29,30]. In contrast, prolonged and recurrent exposure to a high tumor burden in relapsed/refractory cancer patients leads to an exhausted phenotype of T cells and subsequently to an impaired CAR-T cell product [31]. 

CAR-T cell functions consist of cytotoxic capability and immune modulation. The immune synapse is a crucial part of the cytotoxic effect of CAR-T cells. It leads to degranulation and to the release of lytic enzymes such as granzymes, perforin and granulysin. Furthermore CAR-T cells upregulate the expressions of Fas ligand (FasL or CD95-L) and tumor necrosis factor-elated apoptosis-inducing ligand (TRAIL) to generate apoptosis-induced cell death in tumor cells [18]. Furthermore, cytokines and chemokines lead to phenotype changes in T cells and to the recruitment and activation of further immune cells, which in sum can modulate the anti-tumor immune response [18]. 

## 3. Natural Killer Cells

NK cells belong to the innate immune system and are a part of the innate lymphoid cell (ILC) family [32]. They possess a lymphoid character but do not express specific antigen receptors [32]. Because of this, they were identified in the 1970s as killer cells, which, in contrast to the then-known B- and T-Lymphocytes, do not need prior immunization to recognize and kill cells, but rather kill spontaneously [33,34,35]. Rolf Kiessling coined the term natural killer cells in 1975 [36,37]. NK cells possess a repertoire of activating and inhibitory receptors and recognize expression levels of cell surface proteins on target cells. These differing expression levels can signal a “dysregulated self” in the cell [32]. One of the “dysregulated self” signals is the “stress-induced self”, which expresses surface proteins recognised by activating NK cell receptors. The “missing self” state refers to a lack of a constitutively expressed surface protein, e.g., major histocompatibility complex class I (MHC I), which usually inhibits NK cell activation. The cause of these states could be microbial infections, metabolic stress, or malignant transformation [32,38]. Each of these states induces an up- or downregulation of surface proteins, providing NK cells with a surface pattern to recognize. This pattern can either activate or inhibit the NK cells. Activated cells utilize their toolset to interact with the environment or other immune cells, and for target cell elimination. Those mechanisms will be described below and will be summarized in Table 1.

### 3.1. Activation by the Chimeric Antigen Receptor

NK cells have been modified with a CAR to redirect and focus their innate efficient killing mechanisms on specific target cells. The first CAR-NK cell was constructed in 1995 when the ligand binding domain of CD4 was fused with a ζ-chain signaling domain and introduced into NK cells to target HIV-infected T cells or gp120-expressing tumor cells [39]. Antigen binding induced the tyrosine phosphorylation of the ζ-domain. In this context, chimeric receptor-activated signaling pathways acted analogously to FcγRIII/ζ (CD16/ζ) signaling native to NK cells [39]. 

NK cells express several signaling adaptor polypeptides, such as CD3ζ, FcεRIγ (FcRγ), DAP12 or DAP10 (Table 1). These signaling adaptors associate non-covalently with transmembrane receptors upon their activation, e.g., CD16 couples to CD3ζ and FcRγ [40], and are responsible for intracellular signal transduction. CD3ζ, FcRγ and DAP12 contain ITAM, which upon phosphorylation mediates signaling via Syk/ZAP 70 [40,41]. These tyrosine kinases start a cascade that leads to the activation of downstream molecules such as phospholipase C (PLCγ), phosphatidylinositol-3-OH kinase (PI3K), and Vav2/3 (Figure 1). The intracellular domain of DAP10 contains a YxxM motif (x standing for any amino acid), and its signaling is independent of Syk/ZAP70. DAP10 recruits the Grb2-Vav1-SOS1 signaling complex, or can activate the PI3K-Akt pathway (Figure 1) [41]. 

PI3K and Vav recruit the small G protein Rac1 to induce actin polymerization and activate the MAPK signaling pathway through PAK1-MEK-ERK cascade phosphorylation [40,41]. Both ITAM- and DAP10-signaling seem to converge on the activation of Rac1 [40]. Rac1 promotes reorganisation of the cytoskeleton which leads to degranulation that exposes CD107a (LAMP-1) and release of lytic enzymes such as perforin and granzyme [42,43]. The ITAM-signaling pathway additionally activates PLCγ, which hydrolyzes phosphatidylinositol-4,5-bisphosphate (PIP2) into inositol-1,4,5-trisphosphate (IP3) and diacylglycerol (DAG). This causes calcium (Ca^2+^) influx and the translocation of transcription factors into the nucleus, which induces the gene transcription of cytokines and chemokines (Figure 1) [43].

CAR-NK constructs have mostly been fused with a CD3ζ intracellular signaling domain for activation, as this architecture conveniently allows use in both NK and T cells. Only a few studies have used FcRγ, DAP12 or DAP10 as signaling domains [44,45,46,47]. Comparing the signaling domains, Imai et al. have shown the inferior cytotoxicity of peripheral blood NK cells (PB-NK) transduced with an anti-CD19-DAP10 construct compared to an anti-CD19-CD3ζ CAR [44]. For DAP12, the prostate stem cell antigen (PSCA) was chosen as the target and the construct anti-PSCA-DAP12 was compared with anti-PSCA-CD3ζ in YTS NK cells, showing equal IFN-γ-release but a slightly better cytolytic performance of DAP12, especially in lower effector to target (E:T) ratios of 2.5 [47].

Similar to CAR-T cell engineering, co-stimulatory domains have been integrated into NK CAR constructs to boost signal persistence, proliferation, and cytotoxicity. Variations of a co-stimulatory domain including CD28, 4-1BB (CD137), DAP10, and 2B4 (CD244), either individually or in combination, such as CD137-CD28-CD3ζ [48], have produced diverse and sometimes controversial results. Testing CD137-CD3ζ, CD28-CD3ζ, and CD3ζ-containing constructs in NK-92 cells demonstrated that CD137-CD3ζ-CAR-NKs were less effective in killing CD19-positive B-ALL blasts and secreting cytokines, including granulocyte–macrophage colony-stimulating factor (GM-CSF) and IFN-γ compared to the other two constructs [49]. On the contrary, NK cells derived from peripheral blood showed increased cytotoxicity and cytokine secretion of CD137-CD3ζ CAR compared to CD3ζ alone [44]. The study of nine CAR constructs with different co-stimulatory domains in NK-92 or induced pluripotent stem cell (iPSC)-NK cells demonstrated that scFv-NKG2D-2B4-CD3z was the most efficient compared to the third-generation T cell CAR (scFv-CD28-CD137-CD3ζ). It showed the greatest increase in anti-tumor activity and expression of CD107a and IFN-γ, when stimulated [50]. When directly compared, 2B4-CD3ζ demonstrated equal performance to CD137-CD3ζ CARs in PB-NK [51]. Additionally, second-generation CARs with a single co-stimulatory domain performed as well as the third-generation CARs with two co-stimulatory domains [52], raising questions about the effectiveness of using the NKG2D transmembrane domain as an effective component in CAR construction and the importance of tailoring the CAR architecture to the cell type of interest for the best performance.

### 3.2. Kinetics

The kinetic sections will only present data from confirmed cell detection at the specified time point. Persistence beyond this time point is possible but speculative.

Regarding expansion kinetics, the source of NK cells significantly impacts the results. NK cells may be derived from cell lines, which require irradiation before in vivo injection, or from primary cells sourced from peripheral blood, cord blood or stem cells (Figure 2) [53].

NK-92, YTS and KHYG-1 are continuous NK cell lines derived from patients with non-Hodgkin lymphoma, acute lymphoblastic leukemia and NK cell leukemia, respectively. These cells are cultured ex vivo and are transduced with the CAR construct. Upon treatment and before administration, the cell lines are irradiated with 5–150 Gy to prevent uncontrollable proliferation in the host [54,70,71,72].

CAR-NK-92 cells irradiated with 5 Gy or 10 Gy lost their ability to proliferate, with cell numbers decreasing gradually until no viable cells remained at day 7 and day 5, respectively, in vitro [72]. After irradiation, the cells retained their ability for target-specific cell-killing transiently in vitro and in the NSG mouse model in vivo [72,73]. In clinical trials, irradiated NK-92 cells without a CAR construct were detectable in two female patients at 24 h and 48 h after infusion [70]. In the first-in-human clinical trial, irradiated CAR-NK-92 cells were detectable 3 days after their infusion in two acute myeloid leukemia (AML) patients (NCT02944162, phase I trial of CD33-CAR NK-92 cells in three patients with RR-AML) [55]. 

For YT cells, irradiation ranging from 20 Gy to 150 Gy has been described. With 50 Gy irradiation, the percentage of viable cells in vitro decreased rapidly to less than 2% within 6 days, and no viable cells were detectable after 12 days [54]. The capacity for cell lysis after irradiation was retained in vitro until day 5 and confirmed in vivo in NOD/SCID mice, where the CAR-NK cells inhibited tumor growth [54].

KHG-1 cells were irradiated from 1 Gy to 50 Gy, with complete proliferation inhibition achieved at doses of 10 Gy and higher [56]. In vitro, cell viability rapidly decreased, with 89% of cells non-viable on day 3 after 10 Gy irradiation. Cell lysis function was documented until day 2, with a considerable reduction in the cytolytic effect after irradiation (50% cell lysis compared to 89% lysis with the unirradiated control group) on day 2 [56]. Due to these limitations, the impracticality of these cells is underscored by the lack of documented in vivo titres.

For primary-derived NK cells, irradiation does not have to be applied before treatment, and cells are isolated from peripheral blood (autologous and allogenic) or cord blood (HLA-matched or mismatched) and then expanded. Alternatively, NK cells are differentiated from stem cells coming from human embryonic stem cells (hESC), induced pluripotent stem cells (iPSC), human embryos, bone marrow biopsies or leukapheresis after G-CSF mobilization [53]. 

NK cells from peripheral blood (PB) are CD56+ isolated and expanded using different cytokines, primarily IL-2, or with irradiated feeder cells, such as genetically modified K562 cells or autologous peripheral blood mononuclear cells (PBMCs) [74,75,76]. In vitro expansion with IL-2 or IL-15 is slower than with feeder cells, and achieves up to 30-fold expansion in 12 days [77,78] or 1000-fold expansion within 12 weeks [74], whereas activation with gene-modified K562 feeder cells reaches a 1000-fold expansion within 8–9 days [79]. In vitro, peripheral blood anti-CD19 CAR-NKs expanded until day 7 and gradually decreased over time with 20% of cells viable on day 28 [80]. In vivo, the anti-CD20 CAR-NK cells were still detectable on day 11 in NSG mice. The cells were able to distribute into the lung, liver, and spleen [57]. CAR-NK cells, modified with inducible MyD88/CD40 (iMC), sustained expansion and persistence up to day 53 in vivo [58].

Umbilical cord blood (UCB)-derived cells are either CD56+ isolated and expanded in vitro with a cytokine mix or with feeder cells [81], or CD34+ selected and then cultured and differentiated into NK cells with feeder cells within 14 days [82]. In vitro, UCB-CAR-NK cells without additional IL-15 expression exhibited limited persistence, decreasing continually and lasting only around 21 days [83]. In vivo studies in NSG mice showed similar data for UCB-CAR-NK without IL-15 expression, which were hardly detectable on day 21 [83]. In clinical trials, the expansion of UCB-NK cells without any genetic modifications peaked at day 6–8 and decreased further with chimerism detectable up to 14 days [59,60,61]. For one patient donor, UCB-NK cells were detectable up to day 26 [60]. To enhance their persistence, UCB-NK cells have been modified to express IL-15. This additional modification extended the persistence of the UCB-CAR-NK cells, which were detectable for up to 42 days in vitro and up to 68 days in vivo in NSG mice [83]. In a clinical trial, cells expanded between days 3 and 14 and could be detected for at least 12 months in the patients’ blood (NCT03056339, phase 1/2 trial of HLA-mismatched anti-CD19 UCB-CAR-NK in 11 patients with relapsed or refractory CD19-positive cancers) [62]. Another trial confirmed the persistence of CAR-NK cells with IL-15 expression for at least 12 months for responders (NCT03056339, phase 1/2 trial of anti-CD19 UCB-NK/IL-15 in 37 patients with CD19+ B cell malignancies) [63]. In non-responders, the number of CAR-NK cells decreased rapidly around day 42 with no difference in persistence in HLA-mismatch situations [63]. Additionally, anti-CD19-CAR/IL-15-modified UCB-CAR-NK cells showed increased tissue distribution into bone marrow, lymph nodes, liver and spleen compared to CAR-NK cells without IL-15 expression in vivo [83].

iPSCs are preferably used as a source of NK cells due to their homogenous cell population. The differentiation and expansion take up to 3–5 weeks and are slower than for the PB- or UCB-derived NK cells [53,84]. Here, cells are first differentiated into hematopoietic progenitors (CD34+/CD45+) and then into NK cells [84,85]. These cells are termed iNK cells. In vivo, anti-CD-19 CAR-iNK cells persisted for at least for 28 days in NSG mice, and were able to disseminate into the spleen and bone marrow [64]. With an additional IL-15/IL-15R fusion protein gene modification (iDuo CAR-NK), their persistence could be significantly increased, and these cells were detectable throughout day 28 and then gradually declined until day 35 [64]. The number of iDuo effector cells was also elevated in the bone marrow and spleen [86]. iNKs have also been tested in a clinical trial, but the kinetics data have not been released yet [87]. 

### 3.3. Functions

The activation of the CAR triggers various functions in NK cells, which are well documented. One of the core functions of CAR-NK cells is cytotoxicity. Arming with CAR endows NK cells with the ability to recognise NK cell-resistant cells if the corresponding antigen is expressed on the surface. Antigen recognition by CAR activates NK cells and initiates cytolysis. Anti-ErbB2 CAR-NK-92 cells have shown enhanced cytolytic capabilities compared to the parental non-modified NK-92 cell [88,89]. As another function, the NK cells can recruit and interact with other immune cells such as macrophages, as well as T, B and dendritic cells (DCs), and can lead to cell maturation or activation in the process, which in turn contributes to a full-fledged immune response [89]. 

These functions of CAR-NKs result from effector molecules that can be divided into soluble effector molecules and membrane-bound effectors, and will be elaborated in this section and summarised in Table 1. It is important to notice that cell lines and various sources for primary cell-derived NK cells have different characteristics. For instance, NK-92 cells do not express CD16 for antibody-dependent cellular cytotoxicity (ADCC) [90], KHYG-1 hardly express any granzyme B but express more granzyme M than NK-92 and YTS [91], and YTS are not IL-2 dependent [54,92]. Due to these peculiarities, not all effectors mentioned here would apply to all NK cell types.

#### 3.3.1. Soluble Effector Molecules

Soluble effectors can be divided into cytolytic molecules that induce killing mechanisms, cytokines, and chemokines. 

##### Cytolytic Granules

After NK cells bind to their target, cytolytic molecules are secreted from granules at the interface between the two cells. This process is called directed secretion, and requires an immunological synapse (IS) between NK and tumor cells [93]. The immunological synapse is a rearrangement and accumulation of molecules in an immune cell at the junction with another cell. It involves various components such as receptors, signaling molecules, cytoskeletal elements, and cellular organelles [93]. During synapse formation, lytic granules converge to a randomly located microtubule organizing center (MTOC). Together, the MTOC and associated lytic granules polarize to the IS where the granules pass through the actin network and fuse with the cell membrane for secretion [94,95], releasing the main soluble effector proteins perforin, granzyme and granulysin [94,95]. Confocal microscopy reveals such ISs in CAR-NK cells with increased accumulation of CARs, granules and the MTOC polarization at the junction site as the mechanistic basis for the secretion of cytolytic effectors [83]. Only two to four granules are needed to induce cell death in a single target cell, although approximately 10% of the total granule reserve (around 200 granules per cell) is released with each encounter [96]. 

Perforin is a monomeric protein, which oligomerizes at neutral pH in the presence of Ca^2+^ and translocates into the membrane to build membrane-spanning arcs and pores, enabling granzymes to enter the cytosol of the target cell and initiate cell death (Figure 3) [95,97]. The lytic capacity of the perforin pores to induce influx and osmotic lysis seems to be of inferior importance due to the very early onset of the counteracting exocytic membrane repair mechanisms [95]. Therefore, the main function of the perforin pore is to act synergistically with granzymes by ensuring an entry point and sufficient time for the proteases to enter the cytosol [95]. Perforin secretion has been documented for several NK cell sources upon CAR activation in vitro [83,98].

Human granzymes (Gzm) consist of five homologous serine proteases, GzmA, GzmB, GzmH, GzmK and GzmM, and display different mechanisms of cell death induction [99]. GzmA and GzmB are the most abundantly expressed, whereas GzmH/K/M are termed orphan granzymes [100]. These orphan enzymes are regarded as a backup in case the main granzymes are not expressed. NK cells were still able to induce a slow death of virus-infected cells or tumor cells in the absence of GzmA and GzmB, most likely induced by orphan granzymes.

GzmA and GzmK seem to have similar mechanisms and induce caspase-independent cell death through mitochondrial dysfunction and DNA damage without cytochrome c release. They cleave components of the electron transport chain complex I, disrupting mitochondrial redox function and generating superoxides [99]. This oxidative stress translocates the SET complex into the nucleus. The SET complex is a DNA repair complex consisting of DNAses (APE1, NM23-H1, TREX1), the base excision enzyme (BER) and a DNA binding protein (HMGB2) [99,101]. GzmA cleaves the endonuclease inhibitory protein SET, activating both NM23-H1 and TREX1, leading to DNA fragmentation. It also interferes with DNA repair mechanisms by disabling DNA damage recognition and BER. Additionally, GzmA can disrupt chromatin and the nuclear envelope by cleaving histones and lamins, thereby exposing DNA [99]. Compared to GzmB, cell death induced by GzmA occurs more slowly and requires higher concentrations [95]. 

GzmB induces apoptosis through a caspase-dependent pathway, primarily by activating caspase-3 (Figure 3). Caspase-3 then activates the DNAse CAD, resulting in DNA fragmentation [99]. Furthermore, GzmB can cleave ICAD, the inhibitory protein of CAD, and indirectly activate CAD. It can also activate the intrinsic pathway by cleaving Bid, a pro-apoptotic protein, which translocates into the mitochondria after truncation and leads to increased reactive oxygen species (ROS) production, the loss of mitochondrial transmembrane potential (Δψm) and outer membrane permeabilization (MOMP). These changes cause cytochrome c release and initiate the caspase cascade, converging on caspase-3 (Figure 3) [102,103].

GzmH is similar to GzmB because it shares 70% amino acid homology with the latter and is located in the same gene cluster. GzmH causes cell death by mitochondrial depolarization, ROS generation, DNA degradation, and chromatin condensation [99,104,105]. The exact mechanism of cell death induction, including whether caspases, cytochrome c release, Bid activation or ICAD cleavage are involved, remains unclear. Contradictory publications both support and refute the involvement of these proteins [104,105]. 

For GzmM, various studies also state different outcomes and mechanisms. It is consistently stated that GzmM mediates the activation of the adaptor protein FAS-associated death domain (FADD) in the extrinsic apoptotic FAS-FASL pathway. This leads to caspase-8 activation, subsequent caspase-3 activation, and apoptosis [106]. Topoisomerase IIα is another identified substrate of GzmM, whose deactivation leads to cell cycle arrest with ensuing apoptosis [106]. 

The cytotoxic function of orphan granzymes has been explored with recombinant granzymes produced in various organisms, such as bacteria and yeast. This may explain the variations across studies and different, sometimes even contradictory, explanations of the mechanisms of cell death. These assays were mainly performed in vitro, so the actual relevance of cell proteolysis efficacy or cell death induction by orphan granzymes in living organisms is still unknown. Therefore, transferring in vitro results to in vivo or human contexts should be done with caution. At this point, actual relevance for living organisms has only been proven for GzmA and GzmB [107].

Since granzymes were measured at low quantities in healthy human blood samples but showed increased levels during infection or inflammation, it is hypothesized that these enzymes have extracellular functions. GzmA and B have been linked to the degradation of extracellular matrix (such as fibronectin and proteoglycans) and inflammation, with GzmA specifically involved in the cleavage of pro-IL-1β [95,108,109]. GzmM has been linked to immune and blood coagulation regulation. GzmM can cleave the von Willebrand factor and GzmM-deficient mice have decreased serum levels of proinflammatory cytokines such as IL-1α, IL-1β, TNF, and IFN-γ following the LPS challenge [106]. However, the actual relevance of these extracellular functions is still under investigation.

The expression levels of GzmA/B/H/M have been documented for NK cell lines and primary NK cells [91,98,105,110]. GzmK expression depends on the NK subset, being almost completely absent in CD56^dim^ PB-NK cells but present in CD56^high^ cells [110,111]. In the context of CAR activation, only GzmA and GzmB have been measured and reported to be released following antigen stimulation in vitro [98]. CAR-NK-92 cells secrete more GzmA, while peripheral blood NK cells produce more granulysin [98]. Although orphan granzymes have not been investigated for release after CAR-NK cell activation, they can still contribute to tumor cell lysis, as they are known to be expressed and released by various NK cell sources. The extracellular function of granzymes in the context of CAR-NK cells has not been investigated yet.

Granulysin is co-localized with perforin and granzyme in the granules, and is released with these cytolytic effectors (Figure 3). Granulysin is a saposin-like protein, and its main preserved function is to lyse intracellular microbes after entering the cell membrane with or without the help of perforin. Its other function is the cytolytic reaction against tumor cells. The protein can accumulate on the cell surface, lyse the cell membrane, and cause calcium influx [112]. Mitochondrial calcium overload disrupts the redox function and leads to ROS generation and MOMP [113]. This is followed by the opening of megachannels, inducing the release of both cytochrome c and apoptosis-inducing factor (AIF), which activates the caspase cascade and the caspase-independent apoptosis pathway [112,113,114]. Besides cell death, granulysin is a chemoattractant that attracts T cells, monocytes and other NK cells [115]. It also activates immune cells to produce cytokines and chemokines [115]. In the context of CAR-NK cells, its release from the granules after CAR activation has been measured and documented for CAR-NK-92 and CAR-PB-NKs in vitro [98].

The secretion of cytolytic effectors after CAR activation is indirectly measured by CD107a-expression, which is localized in the granules with the effector proteins and released and integrated into the membrane to prevent NK cells from degranulation-associated suicide [116]. Liu et al. demonstrated increased and specific degranulation in CAR-NK cells, compared to non-transduced NK cells, using CD107a flow cytometric measurement when cultivated with CD19-positive tumor cells [83]. CD107a measurements have been provided for several CAR-NK studies for various NK cell sources in vitro, indicating that degranulation is a general mechanism employed by CAR NK cells [51,57,78,117].

##### Cytokines and Chemokines

As part of innate immunity, NK cells possess the ability to modulate the immune response by secreting various cytokines and chemokines (Figure 3). In turn, they can alter protein expression (MHC, Fas, TRAIL, etc.), and recruit and activate other immune cells such as macrophages, T cells and DCs [32]. Upon target cell recognition, NK cells secrete chemokines such as CCL2, CCL3, CCL4, CCL5, CXCL8, and CXCL10 as the earliest response, which peaks at around 1 h and remains stable for at least 12 h [118,119,120]. CCL3, CCL4, and CCL5 are secreted constitutively in a low amount in non-activated NK cells, but with antigen engagement, the secretion can increase up to 181-fold [119]. TNF-α is secreted slightly slower, as it peaks around 3 h after activation and declines again within 12 h [119]. The same applies to IFN-γ, which is secreted even slower, peaks around 6 h after activation and declines within 12 h [119]. Other cytokines and chemokines that are also detected in a very low amount after co-cultivation with K562 are IL-1β, IL-6, IL-7, IL-10, IL-12p40, IFN-α and CXCL9 [119]. The fact that this secretion data does not differ between cytokines from NK cells or the target K562 myeloid cells should be taken into consideration and does not wholly depict specific secretion of NK cells after activation. GM-CSF could hardly be detected alongside the other cytokines and chemokines that are undetectable: IL-2, IL-4, IL-5, IL-13, IL-15, IL-17 and CCL26 [119]. The lack of immunoregulatory cytokines IL-10 or GM-CSF is probably due to the fact that these cytokines are more often reported to be secreted by NK cells upon stimulation with exogenous cytokines (i.e., IL-2, IL-12, IL-15, IL-21) rather than upon antigen recognition [119]. NK cells are one of the first sources of IFN-γ, which is vital for early disease containment before it is provided by activated CD8+ T cells [32]. NK cells also take cues from other immune cells communicated by IL-1, IL-10, IL-12, IL-15, and IL-18. These can, in turn, alter the cytokine response of the activated NK cells, increasing or decreasing cytokine and chemokine responses. NK cells can therefore also work as a relay or an amplifier of signals [119]. 

Imai et al. demonstrated specific cytokine release upon antigen recognition in CAR-NK cells. Anti-CD19-CD3ζ and anti-CD19-CD137-CD3ζ CARs were compared to truncated anti-CD19-CAR without a signaling domain. The comparison showed an antigen-specific CAR-induced release of GM-CSF and IFN-γ caused by the activation of the ζ signaling domain [44]. The cytokine profiles of CAR-NK cells show increments in IFN-γ, GM-CSF, and TNF-α upon CAR activation. These increases have been reported in almost every in vitro study for NK-92, PB-NK, UCB-NK, and iNK cells [44,57,83,121], as well as in vivo [50] and in human clinical trials [63]. In direct comparison, CAR-PB-NKs seem to produce more TNF-α and IFN-γ when compared to CAR-NK-92 cells [98]. In most studies and human clinical trials, IL-6 is only mildly elevated [62,63] compared to the high IL-6 levels seen with CAR-T cells in vivo [50], which explains the low rate of cytokine release syndrome (CRS) adverse events caused by CAR-NK cells [50]. Most studies describe cytokine profiles with hardly detectable IL-6, IL-2, and IL-1β with CAR-NK-92 cell application, as they do not prime monocytes to secrete IL-6 in comparison to CAR-T cells [121]. Chemokine MIP-1α (CCL3) is released after CAR-NK activation in vitro and MCP-1 (CCL2) secretion is detected in a clinical trial on day 3 [62,122]. In this clinical trial, CCL3, CCL5 and CXCL8 are also measured in patient serum, but the results are not further commented on or statistically analyzed, giving no information on altered secretion caused by CAR activation [62].

#### 3.3.2. Membrane-Bound Effector Molecules

NK cells express an array of activating and inhibitory transmembrane receptors, which can be further divided into HLA-specific or HLA-unspecific receptors [61]. The activating HLA-specific receptors include KIR2DS1, KIR2DS2/3, KIR2DL4, KIR2DS4, KIR2DS5, KIR3DS1 and NKG2C, and the activating HLA-unspecific receptors are NKp30, NKp46, NKp44, NKG2D and CD16 [61]. The inhibitory HLA-specific receptors are NKG2A, KIR2DL1, KIR2DL2/3, KIR2DL5, KIR3DL1, KIR3DL2, ILT2/LIR-1 and LAG-3, and the inhibitory HLA-unspecific receptors include PD-1, Siglec-7, TIGIT, TIM-3, Tactile and IL1R8 [61]. CD59, NTB-A, NKp80, DNAM-1 (CD226) and 2B4 have been reported as activating co- and HLA-unspecific receptors [61,123,124]. 

MHC I expression on tumor cells can hamper the cytotoxic function of autologous CAR-NKs, as they can bind to the inhibiting receptor and reduce the killing efficiency. Liu et al. tested the cytotoxic effects of autologous NK cells with or without CAR on CLL cells with high expression of HLA-E [83]. HLA-E binds to the inhibiting receptor NKG2A on NK cells [125]. The autologous CAR-NK cells were able to kill more efficiently than non-transduced NK cells in higher E:T ratios (10:1), but the effect abated with lower E:T ratios (1:1) [83]. Adding an NKG2A inhibitor improved the effect, indicating its involvement [83]. In comparison, cord blood or mismatched CAR-NK cells were more efficient in killing in the presence of the inhibiting MHC I ligand [83]. CAR-NK-92 cells, which only express a few inhibitory receptors, can be of advantage, as they do not seem to be affected by the inhibitory effect of MHC I [89]. HLA-G, which binds to the inhibitory receptor KIR2DL4 on NK-92 cells, was not able to alter the cytotoxic effect of CAR-NK-92 cells [89]. Zhuang et al. engineered a CD28-2B4-CD3ζ, which was able to overcome the inhibiting effects of HLA-E and HLA-C on NKG2A and KIR2DL1 receptors by adding 2BA (CD244) as a co-stimulatory domain, and the CAR-NKs were able to lyse the HLA-E- and HLA-C-overexpressing cell line 221 [126].

Among the activating receptors, CD16 itself can induce targeted cell-killing mediated by antibody-dependent cellular cytotoxicity (ADCC). Except for NK-92 and YTS, all NK sources express CD16 to a higher or lower extent independently of CAR expression [122,127]. CD16 can therefore contribute to the CAR-independent cytotoxic effects of CAR-NK cells [128]. CD16 recognizes the Fc region of immunoglobulin G (IgG), bound to an antigen on the cell surface. Upon activation, CD16 signals through the signaling adaptor polypeptide FcεRIγ and CD3ζ, which transduce their signals through an ITAM and induce cytolysis through Ca^2+^ influx, granule secretion and cytokine and chemokine release [129]. How much CD16 contributes to cytolysis in CAR-NK cells or if the signal from CD16 is even considered and processed within the cell after CAR activation is uncertain.

Apart from these effector molecules, NK cells can reinforce their cytotoxic potential by expressing Fas ligand (FasL or CD95-L) and tumor necrosis factor-related apoptosis-inducing ligand (TRAIL) to induce caspase-dependent apoptosis upon binding to the respective receptors Fas and Death receptor 4 (DR4)/DR5 on target cells [130]. All three receptors belong to the TNF-R family and can trigger apoptosis through their death domain [131]. Cytotoxic effects enacted through FasL and TRAIL, and the activation of the death receptor pathway, are more delayed and may be more relevant in serial killing conditions (subsequent tumor cell cytolysis) beyond initial degranulation upon first tumor cell encounters. Over time, with continuous target cell contact, intracellular GzmB and perforin are reduced, and Fas-L expression increases, indicating the switch from fast-acting granules to slower death pathway-induced apoptosis [132]. The Fas-FasL pathway is induced by the activation of the death domain, which leads to the assembly of the intracellular death-inducing signaling complex (DISC) consisting of FAS-associated death domain protein (FADD), procaspase-8, the modulator FLIP (FADD-like interleukin-1β–converting enzyme (FLICE)–like inhibitory protein), and, sometimes, caspase-10 [131,133]. FADD possesses an N-terminal death effector domain (DED) and binds to the death domain of Fas. The DED of FADD interacts with the DED of procaspase-8 and is then activated into caspase-8, inducing the caspase-dependent pathway [131,133]. The TRAIL-DR4/DR5 pathway starts with the binding of TRAIL on the DR4 or DR5, which also activates the death domain and recruits DISC with a subsequent activation of procaspase-8 into caspase-8 [134]. Prager et al. have shown a notable upregulation of FasL and TRAIL to a lesser extent in NK cells upon co-cultivation with HeLA target cells with ensuing increasing levels of caspase-8 [132]. As for CAR-NK cells, studies have not been as elaborate, as for unmodified NK cells but there have been a few measurements concerning TRAIL and FasL at least indicating that they were expressed on the CAR-NK surface upon target cell encounter [135,136].

#### 3.3.3. Interaction with Other Cells

NK cells can influence other immune cells and vice versa. The main signal receivers upon NK cell activation are macrophages, DCs and T cells. Most of the interaction is induced through cytokines or chemokines. 

IFN-γ is one of the most prominent cytokines, which can activate macrophages and enhance MHC I and II expression on DC surface [32]. It can also increase the loading of peptides onto MHC and antigen peptide production by inducing the expression of proteasome-activator complex PA28, which in turn modulates and enhances the adaptive immune response [32,118,137].

NK cells can recruit conventional DCs (cDC), induce their maturation, increase antigen presentation and perform DC-editing [137]. Dendritic cells are recruited by CCL5, secreted by the activated NK cells, and induced into maturation by IFN-γ and TNF-α [89,137]. Activated NK cells can kill immature DCs by the interaction with NKp30 [137,138]. Mature DCs express a higher amount of MHC I, protecting them from being edited by activated NK cells. The mature DC population is selected by editing to increase successful T cell priming by antigen-presenting cells [137,139]. Mature DCs in turn produce IL-12, IL-15 and IL-18, enhancing the cytotoxicity and IFN-γ secretion of NK cells [32,137,139]. Apart from cytokine and chemokine communication, NK cells can induce up to 21% of non-apoptotic cell killing, indicating necroptosis or pyroptosis [132]. DAMP exposure, which ensues with non-apoptotic cell killing in turn, is recognized by DCs inducing antigen uptake and presentation, fuelling adaptive T cell response [139,140].

Depending on the TME, macrophages can have immunosuppressive as well as activating effects on NK cells [141]. The effect is dependent on soluble molecules such as IL-12 (activating) or TGF-β (inhibiting), as well as direct cell-to-cell interaction with, e.g., NKG2A (inhibiting) or 2B4 (activating) [141]. Primed macrophages can secrete IL-12/-15/-18 to activate NK cells, and accordingly, activated NK cells secrete IFN-γ, TNF-α and GM-CSF to induce a positive feedback loop, and are activated by CD48 or ICAM-1 on macrophages [141].

In the context of CAR-NK cells, the actual role of interactions with other immune cells such as M1 or M2 macrophages, and the possible modulation or repolarization of these cells, as well as DC interactions, still have to be determined. 

## 4. Macrophages

Macrophages are cells of the innate immune system and belong, together with granulocytes and dendritic cells, to the group of phagocytes. Macrophages perform different functions, such as the phagocytosis of invading microorganisms or killed cells, the modulation of immune response by recruiting immune cells, and activating the adaptive immune system or complement system [32,142]. Macrophages are either generated in the bone marrow from common myeloid progenitor cells, which develop into monocytes and differentiate into macrophages after their extravasation into the tissue, or are embryonic-derived with self-renewal and proliferation capacity [32,142]. Most of the tissue-resident macrophages (TRM) in healthy tissue are embryonic-derived and reinforced by recruited myeloid-derived macrophages from the blood if needed [142].

Macrophages are the most abundant immune cells in tumor tissue and TME, which range between 30 and 50% depending on the tumor entity and are defined as tumor-associated macrophages (TAM) [143,144]. In contrast to healthy tissue, TAMs are mostly recruited from bone marrow-derived monocytes, which indicates the high infiltration ability of macrophages into tumor tissue and TME, where other immune cells such as T cells show difficulties in access [142]. Triggered by signals of the tumor or TME, TAMs can differentiate into two distinct phenotypes: M1 macrophages, which exhibit inflammatory and anti-tumor properties, and M2 macrophages, known for their immune-suppressive and pro-tumorigenic characteristics [142,143]. Adapting to changes in the environment, macrophages can dynamically modify their phenotype [142,143]. Polarization to M1 is driven by IFN-γ, TNF-α, GM-CSF, IL-12 and TLR agonists like LPS, whereas M2 polarization occurs under the influence of M-CSF (colony stimulating factor 1, CSF-1), IL-4, IL-5, IL-10, IL-13, prostaglandin E_2_ (PGE2) and TGF-β [142,145]. M1 expresses CD80, CD86, MHC II, and inducible nitric oxide synthase (iNOS), and possesses increased anti-tumor, pro-inflammatory chemokine and cytokine secretion abilities, as well as an upregulated reactive oxygen species (ROS) production and antigen presentation [143,146]. M2 macrophages express CD206, CD204, vascular endothelial growth factor (VEGF), CD163, and arginase-1 (Arg-1), and induce anti-inflammatory, tissue repair and remodelling pathways [143,145,146,147]. 

TAMs can shift their initial phenotype from M1 to M2 through the course of tumor progression. They are involved in tumor progression by promoting angiogenesis through VEGF, immune suppression via TGF-β and metastasis through matrix metalloproteinase (MMP) secretion [148,149]. A high M2 density is associated with a poor prognosis, and therefore TAMs became a key target for tumor therapy [144,150]. TAM-targeting therapies have evolved and include the promotion of phagocytosis, TAM depletion, blocking TAM recruitment, TAM reprogramming and TME remodeling [144]. Building on the TAM-targeting strategy, researchers developed the approach of leveraging the unique capabilities of macrophages themselves. By genetically modifying macrophages and equipping them with a CAR, these cells are redirected and repolarized for enhanced phagocytosis, tumor trafficking, and infiltration. Establishing CAR-macrophages (CAR-M) exploits their abilities of immune modulation, extracellular matrix (ECM) remodeling, and antigen presentation.

Importantly, there is no unified definition of TAMs. While most authors consider all macrophages in the TME and tumor tissue as TAMs, capable of polarizing into M1 and M2 phenotypes, others refer exclusively to M2 macrophages as TAMs. Yet other authors have defined TAM as a unique macrophage subgroup with a specific phenotype and with M1 and M2 features [146]. This review refers to the more general definition of TAM, which refers to all macrophages in the TME and tumor tissue with a potential to polarize into M1 or M2 phenotypes.

### 4.1. Activation by the CAR

The CAR structure of CAR-M is similar to that of CAR-T and consists of an antigen-binding domain, a hinge, a transmembrane region that mostly consists of CD8α, CD147, or CD28, and an intracellular domain, which can differ from other CARs [143]. The intracellular domain can comprise a first-generation CAR with one stimulatory domain or include co-stimulatory domains. The intracellular domain consists mostly of CD3ζ, Megf10, OX40, CD28, CD137, CD86, CD147, Toll/interleukin-1 receptor (TIR), Bai1, PI3K recruitment domain, MerTK, MYD88, and FcRγ (Table 1) [143]. Each component enhances different macrophage properties. The first study to publish on a chimeric receptor in a human monocyte was in 2006, in which a CEA-binding domain was fused to CD64 (FcγRI) [151]. In 2018, Morrissey et al. introduced a CAR for phagocytosis (CAR-P) into murine macrophages using CD19- or CD22- antigen-binding domains and Megf10, FcRγ or CD3ζ as an intracellular domain [152]. In 2019, human macrophages were equipped with an anti-HER2-CD147-CAR, which induced an increased expression of matrix metalloproteinases, remodeled ECM and facilitated the infiltration of T cells into the TME [65]. 

Megf10, CD3ζ, and FcRγ transduce signaling through ITAM, which is phosphorylated by SRC family kinases and recruits Syk [152]. Syk leads to the recruitment and activation of PLCγ, Sos through Shc and Grb2, which initiate the Ras/ERK pathway; the recruitment of PI3-kinase (PI3K) generates PIP3, which in turn activates Vav and Akt (Figure 4) [153]. These pathways ultimately result in actin polymerization and cytoskeletal rearrangements, facilitating pseudopod extension and particle engulfment, as well as the generation of ROS and the production of cytokines like IL-1, IL-8, and TNF-α. This occurs through the activation of the transcription factors NFAT and NF-κB [153]. Bai1 is an engulfment receptor of the adhesion G-protein-coupled receptor subfamily. Bai1 induces the internalization of apoptotic cells through the ELMO (engulfment and cell motility)/DOCK (dedicator of cytokinesis)/Rac signaling module [154]. The adaptor protein ELMO binds to the C-terminus of Bai1 and interacts with DOCK180 to activate Rac, which induces actin polymerization and cytoskeletal rearrangement, along with the subsequent engulfment of the target (Figure 4) [154,155]. MerTK belongs to the TAM family of receptor tyrosine kinases and is one of the endocytotic receptors on macrophages [156]. MerTK recognizes apoptotic cells and mainly leads to efferocytosis, which is induced through the PLCγ-Vav1-Rac1 pathway (Figure 4) [157]. However, the TAM family receptors also express ITIM and can interfere with the TIR signaling pathway, reducing type I IFN release [157]. TAM can also inhibit M1 polarization and enhance M2 polarization [157]. The role of MerTK primarily involves removing apoptotic cell debris, which contributes to the inhibition of inflammatory responses and the maintenance of immune homeostasis, thereby preventing autoimmune disease-like symptoms [157]. This intracellular signaling shows superior phagocytotic ability when anti-CCR7-CAR-MerTK is compared to anti-CCR7-CAR-TLRs [158]. However, this does not necessarily mean a superior specific tumor lysis ability, as TLR signaling also activates other macrophage-activating mechanisms [158]. Furthermore, Bai1 and MerTK show inferior phagocytic abilities when compared to Megf10, CD3ζ, and FcγR as intracellular domains in anti-CD19-CAR [152].

TIR is the key signaling domain of the TLR, which induces cytokine release and enhances phagocytotic activity and ROS/RNS (reactive nitrogen species) production [159,160]. TIR interacts with five adaptor proteins, MyD88 (myeloid differentiation primary-response gene 88), MyD88-adaptor-like (MAL), TIR-domain-containing adaptor protein inducing IFN-β (TRIF), TRIF-related adaptor molecule (TRAM) and sterile α- and armadillo-motif-containing protein (SARM) [159]. The main objective of all these adaptor proteins is the activation of NFκB, which enhances the transcription of iNOS (inducible NO synthase), pro-inflammatory cytokines/chemokines (IL-1β, TNF-α, IL-6, CCL2, CCL3, CXCL8), adhesion molecules (ICAM1) and the regulation of proliferation, differentiation and survival [159,160]. MyD88, with the help of MAL, recruits IRAK1 and IRAK4, which leads to the activation of TRAF6. This in turn activates the IKK complex, resulting in the release and translocation of NF-κB (Figure 4). TRAF6 can also interact with IRF1, IRF5 and IRF7, leading to the transcription of TNF or type I IFN, and also links to the p38 and JNK pathway, influencing macrophage pro- or anti-inflammatory response, development, proliferation, and survival [159,161,162]. TRIF interacts with TRAF3 through the additional recruitment of TBK1, which activates IRF3/7 (inflammatory cytokine production), and TRIF interacts with TRAF6, activating the IKK complex and NFκB (Figure 4) [159]. TRIF creates an additional link to FADD, which can lead to apoptosis [159]. SARM is a negative regulator of TRIF and TRAM and is a bridging adaptor to TRIF, but also has individual functions and is needed for the induction of TNF, IL-6, CD86 or IRF3 [159].

Second-generation CARs are in the making; adding the PI3K-recruiting domain to FcRγ increased trogocytosis and tumor cell reduction, but most notably enhanced whole-cell engulfment [152]. By combining the phagocytosis-inducing CD3ζ domain and the pro-inflammatory TIR signaling domain, Lei et al. were able to show improved killing activity, an increased release of pro-inflammatory cytokines and a longer maintenance of M1 polarization with the second-generation CAR-CD3ζ-TIR compared to the first-generation CAR-CD3ζ or CAR-TIR with anti-EGFRvIII/GPC3 as the antigen-binding domain [163]. 

### 4.2. Kinetics

Peripheral blood is the primary source of CAR macrophages, and since HLA matching is not essential when antigen presentation is not a priority, alternative sources such as cell lines (THP-1, U937, SC) and stem cells (from cord blood, bone marrow, and induced pluripotent stem cells) can be utilized [143]. For mouse models, murine bone marrow-derived macrophages and the cell lines J774A.1 and RAW 264.7 have been employed. 

Zhang et al. reported that iPSC-derived macrophages (CAR-iMac) with an anti-CD19 or anti-mesothelin CAR experienced a two-fold expansion by day 3 in NSG mice, which persisted for more than 20 days and then gradually disappeared around day 30 (Figure 2), but without any statement as to which compartment the cells were measured in [66]. Unmodified bone-marrow-derived macrophages from MacGreen mice were used for the treatment of liver fibrosis in C57BL/6 mice; they were detected on day 1 in the liver and showed a gradual reduction within 7 days, and were not detected anymore after 4 weeks [164]. For RAW246.7 cells in Balb/c mice bearing 4T1 orthotopically, the anti-HER2-CAR-M were detected at the tumor site on day 1, with a maximum signal on day 3 [65]. These CAR-M mostly accumulated in the liver and were hardly detectable on day 6 [65]. Klichinsky et al. noted at least 62 days of persistence for PB-CAR-M in tumor-free NSG mice (Figure 2) [67]. In tumor-engrafted NSG mice, CAR-M accumulated at the tumor site and also in physiological tissues such as the liver, lung and spleen, in quantitatively decreasing order, on day 5 [67].

In clinical trials, only primary human autologous macrophages such as PB-CAR-M and human bone marrow stem cells CAR-M have been documented to this point. The kinetics concerning CAR-M in clinical trials have not been released yet. The latest release on CT-0508 (NCT 04660929 human primary anti-HER2-CAR-M) has stated a rapid egress of CT-0508 from peripheral blood and CAR mRNA detection in all tumor biopsies of the first two patients without reference to time points [165]. In screening for unmodified autologous macrophage trials, intravenously injected PB-derived macrophages were retained in the lungs or liver (injection through the hepatic artery) up to 90 min after injection, and accumulated afterwards at the metastatic sites, especially the liver, where they were detectable for more than 7 days [166]. Macrophages, which were injected intraperitoneally, also accumulated around metastatic sites and remained detectable for more than 7 days [166]. A correlation between persistence and outcome is still unclear and needs further investigation, specifically in clinical trials. 

### 4.3. Functions

Macrophages belong to the innate immune system, and apart from pathogen defence, fulfil several other functions, such as clearance of cell debris and apoptotic cells, as well as tissue homeostasis and wound regeneration [160,167].

After CAR activation, macrophages can initiate various functions depending on the specific signaling pathway engaged. Functions are summarised in Table 1.

#### 4.3.1. Soluble Effector Molecules

Upon CAR activation, macrophages can secrete several effector molecules, which include cytokines, chemokines as well as matrix metalloproteases (MMP). Unlike CAR-T and CAR-NK cells, CAR-M can additionally influence and modulate the TME. Secreted cytokines can activate bystander immune cells and induce phenotype changes in TAMs, as well as promoting antigen uptake and the maturation and presentation of DCs, or enhancing the cytolytic functions of NK and T cells. Chemokines recruit further immune cells to the tumor site and MMPs can influence the permeability of TME, facilitating immune cell invasion.

##### Cytokines and Chemokines

Activation of the CAR mostly induces M1 polarization of the macrophage, and therefore the cytokine profile corresponds to an M1 phenotype response. Upon activation, M1 macrophages can secrete cytokines IL-1α, IL-1β, IL-6, IL-12, IL-18, TNF-α and M-CSF and chemokines CCL-2, CCL-3, CCL-4, CXCL-8 and CXCL-10 [160]. 

CAR-M from the cell lines U937, SC, and THP-1 with an OX40-CD28-CD3ζ signaling domain have been shown to express markers for an M1 phenotype and to secrete pro-inflammatory cytokines IL-1β, IL-8 and TNF-α, whereas IL-10 was not secreted in vitro [168]. For stem-cell derived macrophages, the GD2-CAR with the same domains OX40-CD28-CD3ζ additionally showed the secretion of CXCL10 and CCL3 [169]. Lei et al. compared anti-EGFRvIII-CAR with a CD3ζ (CD3ζ-CAR), TIR (TIR-CAR) and second-generation CD3ζ-TIR (CD3ζ-TIR-CAR) signaling domain in iPSC macrophages to each other, showing a significant increase in cytokine secretion in the second-generation CAR compared to the first-generation [163]. Gene expression for IL-1α, IL-1β, IL-6, IL-12, IL-23, CCL8, CXCL8, and TNF-α was significantly higher in the second-generation group, and a higher secretion of IL-6, IL-12, IL-23 and TNF-α was confirmed in the supernatant after 24 h of tumor cell co-cultivation in vitro [163]. In vivo, RAW264.7 were transduced with CCL19-MerTK CAR and injected into 4T1-bearing BALB/c mice [158]. Upon activation, CAR-M showed an elevation of IL-6, CCL2, TNF-α and IL-1β in the serum compared to empty vector CAR-treated mice or naïve untreated mice, whereas IL-12, IL-10, IL-4 and IL-13 remained at the baseline [158]. Both in vitro and in vivo, cytokine release syndrome (CRS)-related cytokines such as TNF-α, IL-6, and CCL2 are secreted by CAR macrophages (CAR-M) at levels showing a 1.5- to 10-fold increase, compared to the 30- to 8000-fold increase observed after CAR-T cell injection. This suggests a lower likelihood of inducing CRS with CAR-M treatment [169].

##### Matrix Metalloprotease (MMP)

In addition to cytokines and chemokines that modulate immune responses and recruit immune cells, macrophages also secrete enzymes known as matrix metalloproteinases (MMPs), which are crucial for remodelling ECM [170]. There are 23 MMPs expressed in humans, and these are classified according to their substrates as collagenases (MMP-1, MMP-8, MMP-13), gelatinases (MMP-2, MMP-9), stromelysins (MMP-3, MMP-10, MMP-11), matrilysins (MMP-7, MMP-26) or membrane-type MMPs (MT-MMPs) [171]. Human macrophages express MMP-1, -2, -7 to -9 and -12, -14, -19, -23 and -25 (Figure 5) [170]. Murine macrophages express similar MMPs except for MMP-7, and show a higher expression of MMP-12, -13 and -23 than human macrophages [170,171]. Activator protein-1 (AP-1) and NF-κB transcription factor pathways upregulate MMP-1, -3, -10, -12 and -14 in human macrophages [171,172]. However, the MMP expression is also modulated by other proximal transcription factor binding sites, such as specificity protein-1 (SP-1) and signal transducer and activator of transcription-1 (STAT-1) [171,172]. This upregulation of MMPs was confirmed in anti-HER2-CD147-CAR in RAW246.7 macrophages, which were injected into Balb/c mice engrafted with 4T1 tumors [65]. CD147 signals through Interferon-induced transmembrane 1 (IFITM1) and activates NF-κB by recruiting the signaling adaptors ERK and PI3K [173]. MMP-3, -11, -13, and -14 were significantly upregulated in CD147-CAR-M. With a higher E:T ratio (2:1), the expressions of MMP-9, -10 and -12 were additionally upregulated [65]. The CD147-CAR did not lead to any change of surface markers except for an increase in CD80. Similarly, no changes in cytokine levels, phagocytosis activity, ROS production in vitro or macrophage infiltration into the tumor in vivo were noted [65]. The activation of the CD147-CAR increased MMP release, facilitated CD3+ T cell accumulation within the tumor tissue and TME, and led to significant tumor control compared to the control group in vivo [65]. In vivo, cytokine measurement showed no sign of CRS, but led to decreases in IFN-γ, TNF-α, and IL-6 levels in the serum instead [65]. In tumor tissue samples of the CAR-M group, an increase in local IL-12 and IFN-γ was measured [65]. Tumor control seems to be related to increased T cell infiltration, as the injection of CD147-CAR into Balb/c nude mice with T cell deficiency did not lead to tumor control [65].

##### ROS, RNS, NO and Other Soluble Molecules

ROS stands for a group of reactive O_2_ derivatives, the most abundant derivatives being O_2_^−^ and H_2_O_2_. With TLR activation and endocytosis of the target material, ROS production is induced by NAPDH-oxidase (Nox) in the phagosome membrane. This enzyme transfers an electron and reduces O_2_ to O_2_^−^, which reacts with H_2_O to form H_2_O_2_. O_2_^−^ (superoxide) can also be produced in the mitochondria during respiratory activity in the electron transport chain (ETC) [174]. O_2_^−^ is highly reactive and cannot cross cell membranes [174]. High amounts of O_2_^−^ lead to the non-specific oxidation of proteins, amino acids, DNA and lipids, impairing their function and inducing oxidative stress with subsequent cell death [174,175]. H_2_O_2_ can diffuse through membranes and, apart from oxidative stress, it can contribute to cell signaling [174]. H_2_O_2_-mediated signaling is mostly based on the oxidation of cysteine residues, and can generate immune-modulatory functions. ROS can covalently link IKK and NEMO via a disulfide bond and activate NF-κB, promoting pro-inflammatory response and cytokine secretion (Figure 5) [174]. By deactivating cathepsin in the phagolysosomes, ROS can prevent excessive proteolysis and enhance antigen presentation on MHC [174]. Also, ROS can lead to NLR3 inflammasome assembly with the subsequent cleavage of pro-IL-1β and pro-IL-18 into their active forms [174]. 

NOS is expressed during pro-inflammatory signalling, and its expression is used as a marker of M1 macrophages. The induced NOS (iNOS) in macrophages produces NO by catalyzing the reaction of arginine and O_2_ to NO and citrulline. NOS can also produce O_2_^−^ when arginine is not available [174]. Just like O_2,_ NO also has reactive derivatives such as peroxynitrite grouped under reactive nitrogen species (RNS) [176]. Depending on their concentration, RNS and NO can induce damage or serve as cell signaling molecules [176]. RNS can cause DNA breaks and mutations, as well as interact with proteins by oxidating heme groups or thiol groups in cysteine and methionine, or it can create covalent bonds, thereby inactivating protein functions [176]. High amounts of RNS can modify death receptor (FAS, TNFR, TNF-α) expression with increased apoptosis, and can inhibit the mitochondrial respiratory chain and cause MOMP with increased cytochrome c release, along with the induction of apoptosis [176]. Nitrated p53 shows enhanced DNA binding and increased apoptosis. But NO/RNS also have pro-tumorigenic effects, especially in lower concentrations. In low amounts, RNS/NO can nitrosylate caspase-3, leading to enzyme inactivation, and nitrosylation can also protect anti-apoptotic proteins (FLIP, Bcl2) from ubiquitination and degradation [176]. The induction of iNOS was confirmed for CAR-M and also for M2-macrophages, which were co-cultivated with CAR-M, causing dynamic phenotype modification [67]. 

ROS and RNS are mainly used to induce oxidative bursts in lysosomes to kill phagocytosed microbes, but due to their intracellular signaling and immune-modulating function, they can enhance the pro-inflammatory responses of the macrophage towards the tumor cell. The release of ROS and RNS with subsequent diffusion into tumor cells is controversial and still under investigation.

Less documented, macrophages have been shown to express GzmA, GzmB, GzmK and GzmM [177,178,179,180,181], but do not seem to express or secrete perforin-1 (perforin of NK and T cells), indicating a more immune-modulatory role of the granzymes than a cytolytic effect [179]. In particular, GzmA seems to enhance TLR4-induced pro-inflammatory cytokine production and can directly cleave intracellular pro-IL-1β [179]. Macrophages do not seem to express perforin-1 but express perforin-2 instead [179,182]. Perforin-2 is located in intracellular vesicles, which fuse with phagolysosomes and form pores in engulfed pathogens within the phagolysosomes, facilitating ROS and RNS-induced pathogen-killing [179,182].

##### Apoptotic, Necroptotic and Pyroptotic Effects of Soluble Effector Molecules

TNF-α can bind to TNFR1 and lead to various pathways, such as NF-κB activation, apoptosis and necroptosis. Depending on the homeostasis of the signaling pathways, this can lead to the proliferation and survival of the cell or cell death [183]. By binding TNF-α, the receptor trimerizes with the release of silencer of death domain (SODD) and TRADD, and RIPK1 can bind to the intracellular death domain of TNFR1 [183,184]. Apoptosis occurs via the recruitment of FADD, and this complex induces caspase-8 activation and apoptosis (Figure 5) [185,186]. Another pathway of TNFR1 signaling is the necroptosis pathway; here, the pathway first leads to complex I assembly and to NF-κB activation, and in a second step leads to complex II grouping and necroptosis (Figure 5). In complex I, TRADD and RIPK1 lead to the binding of TRAF2 and co-recruitment of cIAP, which modifies the TNFR1 complex with ubiquitin chains, a binding site for LUBAC. LUBAC modifies RIPK1, which causes the recruitment of TAK1, TAB2 and IKK complexes, leading to IKK degradation and NF-κB activation [185,186]. NF-κB induces pro-survival signals by the expression of FLIP, which heterodimerizes with caspase-8 and inactivates the protease. Caspase-8 inhibition leads to RIPK1 de-ubiquitination and activation. RIPK1 then interacts with TRADD, FADD, caspase-8 and RIPK3, building complex II [185]. Activated RIPK3 activates MLKL (mixed lineage kinase domain-like protein), which in turn oligomerizes and translocates to the membrane to form pores, or interacts with ion channels, both leading to necroptosis [185,186]. Although TNFR1 can be expressed in tumors, TNF-α signaling has not yet been demonstrated for CAR-M towards tumor cells as a cytotoxic effect. TNF-α also has a controversial role in carcinogenesis. It has shown pro-tumorigenic as well as anti-tumorigenic effects. By activating NF-κB, it can induce MMP production in tumor cells, especially MMP-9, leading to the invasion and dissemination of tumor cells. TNF-α can also induce epithelial–mesenchymal transition (EMT), which can lead to cell adhesion loss and the disruption of cell–cell junctions with increased migration and invasion [187,188]. Also, TNFR1 is expressed on immune cells, and TNF-α can be attained by CD8+-TIL, leading to activation-induced cell death (AICD) [188].

Pyroptosis induction by macrophages has been described for breast cancer cells [189]. Pyroptosis is mainly induced by cleaving gasdermin family proteins (GSDM), which can form pores in the cell membrane and lead to cell lysis. There are several pathways to induce pyroptosis [189]. The classical pathway starts with the activation of the NLRP3 inflammasome, leading to ASC adaptor protein and pro-caspase recruitment with subsequent caspase-1 activation [189]. Capase-1 activates IL-1β and IL-18, and cleaves Gasdermin D (GSDMD). The N-terminal GSDMD part assembles on the cell membrane, binds to phosphatidylinositol phosphate, phosphatidylserine and cardiolipin on the cell membrane, and forms a pore as cytolytic effect [189,190,191]. The non-classical pathway activates caspase-4 and -5, which also activates GSDMD [189,190,192]. Other pathways involve caspase-8 activation and the cleavage of GSDMC, which is activated by TNF-α, or caspase-3 recruitment with subsequent GSDME activation [190]. GzmA can directly activate GSDMB [190]. The triggers for inflammasome assembly or caspase activation are soluble effectors released by macrophages, and they consist of PAMP/DAMP (e.g., HMGB-1), with the activation of pattern recognition receptor (PRR), or TNF-α, with TNFR activation, both leading to NF-κB upregulation; the trigger also include GzmA, GzmB and microRNA-155-3p uptake (Figure 5) [189].

Concerning CAR-M, a TUNEL (terminal deoxynucleotidyl transferase dUTP nick end labeling) test has been run on in vivo tumor samples, and has confirmed the increase in apoptotic cells in the tumor tissue after CAR-M treatment compared to the control group, indicating apoptosis induction. Whether apoptosis was induced by CAR-M or by infiltrating T cells cannot be distinguished [193], and leaves the documentation of apoptosis, necroptosis and pyroptosis induction by CAR-M lacking.

#### 4.3.2. Membrane-Bound Effectors

Macrophages have various receptors used to recognize different cells and pathogens. The receptors can be classified by the binding ligand type, such as unknown molecules detected by their patterns, opsonins, and apoptotic cells [194]. 

Pattern recognition receptors (PRR) recognize PAMPs such as viral single- and double-stranded RNA (ss-, dsRNA), LPS, host heat-shock proteins (HSP) and fungi proteins such as mannan [195]. PRRs consist of various receptor types and families: TLRs (TLR-3, -4, -7, -8, -11 are expressed on macrophages), scavenger receptors (SR-A, MARCO), C-type lectin receptors (CLRs) and NOD-like-receptors (NLR), which are expressed in the cytosol [167,195,196]. 

Opsonins are complement factors (C3b, iC3b) and antibodies (IgG1-4) that bind to the surfaces of pathogens and are recognized by complement receptors (CR1 and CR3) and Fc receptors for IgG (FcγRI-III). The latter ones are used for antibody-dependent cellular phagocytosis (ADCP) [196,197,198]. 

Apoptotic cells are recognized by phosphatidyl serine receptor (PSR), scavenger receptors (CD36, CD14) and vitronectin receptor αvβ3-integrin, which recognizes thrombospondin, the TAM receptors (Tyro3, Axl, MerTK) and several other receptors [167,196].

The intracellular signaling of TLR and NLR activate NF-κB and IRF3, leading to the transcription of pro-inflammatory cytokines and chemokines for immune modulation or caspase activation, with the cleavage of pro-IL-1β, pro-IL-18 and pyroptosis [194]. 

Phagocytotic receptors induce target engulfment and can further be classified into efferocytosis-inducing receptors with anti-inflammatory responses or phagocytosis-inducing receptors with subsequent inflammatory response (Figure 5). Efferocytosis is prompted by receptors that recognize patterns on apoptotic cells, such as phosphatidyl serine (PS) receptors or TAM receptors, which recognize linkers that bind to exposed PS, e.g., growth arrest-specific protein 6 (Gas6), and protein S [194]. TAM receptors can form a complex with the IFN-γ receptor domain R1, inducing STAT1 phosphorylation and translocation to the nucleus as well as the gene expression of suppressor of cytokine signaling (SOCS) 1 and 3. SOCS binds to JAK kinases and causes the proteasomal degradation of associated proteins, or can function as a JAK pseudosubstrate leading to the inhibition of cytokine receptor signaling, the termination of inflammatory responses and immunosuppression [199]. 

Phagocytosis induced by FcγR or CR on the other hand induce engulfment and pro-inflammatory cytokine expression (see activation by the CAR) [200]. 

Therefore, the choices of different receptor signaling domains for CAR engineering should depend on their pathway activation and the intended effect. CARs used for comparison to each other should be matched concerning their effects, as TLRs cannot lead to sufficient phagocytosis compared to MerTK. TLR can only enhance phagocytosis, but cannot induce phagocytosis by phagocytotic receptor signaling, while MerTK cannot induce NO as it has anti-inflammatory qualities [158]. When endocytosis receptors are compared to each other, CARs with phagocytotic receptor ITAM signaling domains seem to be superior to those with efferocytosis receptor signaling domains MerTK or Bai [152].

FasL is expressed in T, NK cells, and macrophages [201,202]. It can bind to Fas (CD95) on target cells to induce apoptosis (see NK cell section). In one study with colon cancer, macrophages on the invasive margin demonstrated increased FasL expression [203]. A positive correlation between macrophage quantity and apoptotic tumor cells in the invasive margin was observed. Additionally, the detection of tumor cells within the macrophage cytoplasm indicates tumor cell killing by FasL apoptosis with subsequent phagocytosis [203]. However, how much FasL contributes to the tumor cell cytotoxicity of macrophages is still unknown and under investigation. Furthermore, most publications consider FasL expression on macrophages as a means of regulation, as it can lead to apoptosis of bystander immune cells such as neutrophils, monocytes and lymphocytes, as part of immune deletion for homeostasis, and the termination of an inflammatory response [191,201,203,204,205]. The role of FasL in CAR-M is still undocumented and unclear.

TRAIL expression has been documented for macrophages, and TRAIL can induce apoptosis and enhance tumor control of colon cancer cells [206]. TRAIL, just like FasL, also seems to have a regulatory function, but only on monocytes, while neutrophils and lymphocytes only express the non-signaling DR and are not affected by TRAIL [206,207]. The role of TRAIL in CAR-M cells is still in need of investigation.

#### 4.3.3. Phagocytosis and Antigen Presentation

Phagocytosis activation leads to actin rearrangement and the formation of a phagocytic cup (depression in the membrane), as well as subsequent pseudopod formation around the target until it is fully engulfed, thereby creating a new phagosome inside the cell [200]. Phagosomes undergo maturation by fusion with early endosomes, late endosomes and lysosomes forming the phagolysosome. During this process, phagosomes acquire different properties and processing and recycling methods, such as proton-pumps for acidification and degrading enzymes (cathepsins, proteases, lysozymes, lipases) to digest the engulfed material [200]. Peptides resulting from this digestion are loaded onto MHC II and are translocated to the membrane for antigen presentation [32]. MHC II is recognized by the TCR of CD4+ T helper cells, activating the adaptive immune system [32]. Cross-presentation to CD8+ cells via MHC I on TAM has been observed by single studies, and induces an earlier response in activated than in naïve T cells due to the lack of co-stimulatory domains. Furthermore, the exact function of this cross-presentation is still unclear, as there are studies that show that CD8+ TILs activated by macrophage cross-presentation exhibit a highly activated phenotype but lack effector cell functions, especially in killing assays in vitro [208,209]. Whether this impairment comes from macrophage cross-presentation or TME’s influence is unclear, and requires further investigation. The recognition of peptides on the MHC I/II by a TCR is HLA-dependent and needs to be HLA-matched.

In the context of CAR-M cells, Klichinsky et al. were able to demonstrate T cell activation and proliferation caused by antigen-presenting CAR-M cells in vitro. Antigen presentation was achieved by transducing HLA-A2 NY-ESO1 into CAR-Ms. This successfully activated the complementary anti-NY-ESO-1 Ig4-TCR-CD8+-T cells, measured as increased proliferation compared to T cells cultivated with the control CAR-M without NY-ESO-1 [67]. The authors were also able to show CAR-M antigen presentation, especially cross-presentation, after the phagocytosis of SKOV3 tumor cells, which were additionally transduced with NY-ESO-1 by measuring the T cell proliferation of Ig4-TCR-CD8+-T cells also in vitro [67]. The demonstration of antigen presentation in vivo was achieved by engrafting SKOV3 in NSG-SGM3 mice and injecting the mice with CAR-M only, CAR-M and polyclonal T cells, or polyclonal T cells only [67]. The combination of CAR-M and polyclonal T cells induced tumor reduction compared to the other two conditions, indicating the presentation and activation of T cells [67].

Considering that HLA needs to be matched for TCR to recognize peptides on MHC, the sources of macrophage differentiation should be considered. Off-the-shelf macrophages, such as cell lines as well as iPSC-derived macrophages, for which an HLA ablation could be employed to reduce graft rejection and to generate a “universal” CAR-M, may inherit inadequate or ablated antigen presentation and deficient antigen recognition [210]. 

#### 4.3.4. TME Remodeling 

Apart from functions derived from the signaling pathways after CAR activation, the mere transduction itself can already polarize CAR-Ms to an M1 phenotype, especially when transduced with an adenoviral vector [67]. Transduction induces interferon signaling, PRR, iNOS signaling and MHC class I/II gene inductions, elevating antigen presentation and indicating an activated M1 phenotype [67]. This polarization was not dependent on CAR expression, as transduction with an empty adenoviral vector led to the same activation pattern as a CAR-containing adenovirus vector [67]. This M1 phenotype can be maintained for at least 40 days after transduction, and can even be sustained within the TME of humanized NSG mice engrafted with ovarian tumor SKOV3 [67]. RNA sequencing of the tumor from a humanized mouse model 5 days after CAR-M injection confirmed two distinct clusters of untransduced (UT)- and CAR-Ms, with the CAR-M cluster showing an increased expression of pro-inflammatory genes (TNF, MHC II), indicating the remodeling effect of CAR-M cells on the TME [67]. Macrophages have been excluded from the RNA sequencing data to depict only TME gene expression [67]. Furthermore, CAR-M induced pro-inflammatory marker expression on M2A, M2C and M2CD macrophages upon co-cultivation, and enhanced T cell activation and chemotaxis [67]. Following in vivo transfection via nanocomplex delivery to macrophages, anti-ALK-CD28-CD3ζ-IFN-γ CAR-M was created in situ intratumorally for subsequent TME remodeling [193]. After 16 days, tumor tissue was processed, and an increase in M1 markers and a decrease in M2 markers, as well as an increment in pro-inflammatory cytokines such as TNF-α and a reduction in immune-suppressing cytokines IL-4, IL-10 and TGF-β, were measured [193]. Furthermore, CAR-M was able to recruit and activate CD8+ T cells and increase GzmB and IFN-γ in the tumor tissue, whereas the regulatory T cell (T_reg_) population decreased significantly [193]. These findings support the hypothesis that the modification of TAM into an M1 macrophage will also affect immune cells in the tumor tissue and TME.

## 5. γδ T Cells

Gamma-delta (γδ) T cells are unconventional, mostly double-negative T cells that express the heterodimeric TCR with γ- and δ-chains. They form a distinct subset separate from single positive CD4+ or CD8+ T cells, which express αβ TCRs [211]. 

γδ T cells differ from αβ T cells in their distribution, recognition receptors, functions and response upon activation. They distribute to peripheral tissue rather than to lymphoid organs, and are MHC-independent [212,213]. Functionally, γδ T cells are known to act like APCs and express various receptors for antigen recognition that are not MHC-dependent, and therefore do not need antigen presentation and show decreased alloreactivity [212]. Due to these special features, as well as their low graft-versus-host-disease (GvHD) side effects, their tissue residency and their rather easy expansion ex vivo compared to NK cells and macrophages, γδ T cells have been increasingly recruited for use in adoptive cell therapy, with a possibility of allogeneic donorship.

γδ T cells are classified by their γ-chains in mice and by δ-chains in humans, with Vγ9/Vδ2+ T cells (Vδ2 T cells) forming the most prominent population in human peripheral blood. Non-Vδ2 T cells contain Vδ1, Vδ3 and Vδ5 and are distributed to peripheral tissue and organs [212]. Due to their easy isolation from peripheral blood and expansion ex vivo by using bisphosphonates (e.g., Zoledronate), most γδ T cell immunotherapy studies focus on the Vδ2 T cell subset [214]. With new expansion protocols for Vδ1 T cells and due to their special features—tissue residency, decreased susceptibility for activation-induced cell death (AICD) and longer persistence—Vδ1 T cells have recently come into focus for adoptive cell therapy [215].

### 5.1. Activation by the CAR

The CAR structure for γδ T cells does not differ from the CAR constructs for other cell types. It consists of an extracellular antigen recognition domain, a hinge region, a transmembrane region and the intracellular (IC) signaling domain, which is mainly composed of CD3ζ (Table 1) [216]. In one study, DAP10 alone was used as a CAR signaling domain to confer increased tumor specificity and decreased on-target off-tumor effects. The corresponding construct was termed co-stimulatory-only chimeric antigen receptor (CCR) [216,217]. DAP10 signaling alone was not able to induce full activation of γδ T cells with cytotoxicity and release of cytokines [217]. Only with simultaneous activation of the Vδ2 T cell TCR, which recognizes phosphoantigens on stressed, infected or tumor cells, did CAR-DAP10 signaling mount a full response with release of IFN-γ, TNF-α, IL-2, IL-4, GzmB and cytotoxicity against tumor cells [217,218]. Anti-GD2-DAP10 CAR conveyed increased specificity towards malignant cells, and was not activated by normal cells without the stress signal [217]. This combination was as effective as the second-generation anti-GD2-CD28-CD3ζ CAR [217]. With the same intention of decreasing on-target off-tumor effects, a CAR without a signaling domain has been created, termed non-signaling CAR (NSCAR). NSCAR equips γδ T cells with enhanced antigen targeting and uses the innate killing activation mechanisms of the cells for tumor cytotoxicity [219]. Anti-CD19-NSCAR transduced into γδ T cells was able to increase target-specific tumor cell killing 1.5-fold compared to UT cells, and depicted minimal cytotoxicity against healthy B-lymphocytes comparable to UTs [220]. The same was observed for anti-CD33-NSCAR when incubated with CD33+ cells isolated from healthy donors: the cytotoxicity was <10%, which is comparable to that of UTs [220].

CAR constructs in γδ T cells utilize CD137, CD27 and CD28 as co-stimulatory domains [216]. Du et al. compared several anti-CD19 CARs, which were expressed in a mixture of cytokine-induced killer cells (CIK) with 20% of Vδ2 T cells [221]. In this study, the CD27-CD3ζ and CD28-CD3ζ CARs showed comparable toxicity and cytokine release, whereas the third-generation CAR containing CD28-CD137-CD3ζ also showed comparable cytotoxicity, but showed decreased cytokine release in comparison to second-generation CARs [221]. All three were superior to the third-generation anti-CD19-CD27-CD28-CD3ζ combination [221].

CD3ζ is part of the TCR complex, and the signal is transmitted by phosphorylating its intracellular ITAMs. αβ TCR consists of α and β TCR subunits and the CD3 subunits ε, γ, δ and ζ [222,223]. αβ TCR initiate three main pathways: calcium flux, NF-κB and Ras-MAPK activation [21]. Upon activation, Lck tyrosine kinase is recruited by CD4 and CD8 co-receptors, which phosphorylates the ITAM region, leading to the association of ZAP70 [223]. ZAP70 activates the Lat signalosome, which is a docking site for adaptor proteins such as Vav1, Sos1, Grb and PLCγ [224]. The Vav1-Rac pathway triggers actin polymerization as well as Fos expression, which in conjunction with Jun forms the transcription factor AP-1 [225]. PLCγ hydrolyzes PIP2 into IP3 and DAG [224]. IP3 induces Ca^2+^ release from the endoplasmic reticulum, the activation of calcineurin and the translocation of transcription factor NFAT into the nucleus, which forms a complex with AP-1 [225]. DAG recruits Ras guanyl-releasing protein 1 (Rasgrp1), which activates the Ras-MAPK-ERK-JUN pathway and protein kinase C (PKC), and subsequently the NF-κB pathway [224]. The AP-1 complex and NF-κB induce phenotype differentiation, survival and homeostasis, as well as cell cycle transcription factors, cytokine, chemokine and effector molecule expression [21,225].

Although αβ TCR signaling is well investigated, little is known about γδ TCR signaling. γδ TCR consists of γ and δ TCR subunits and the CD3 subunits ε, γ, δ and ζ, but it differs in the geometry and arrangement of the CD3 subunits and ITAMs compared to αβ TCR. After activation, γδ TCR also shows differences in the receptor proximal signaling [223,224]. The recruitment of Lck is unclear and needs further investigation, as mostly no CD4- or CD8-co-receptors are available to γδ T cells [216,224]. Furthermore, the survival of γδ T cells in Lck- and ZAP-70-deficient mice while αβ T cells are completely absent implies an alternative activation of the receptor distal signaling pathway [224]. Additionally, CD3ζ-deficient mice seem to be able to substitute CD3ζ for FcRγ domains [223,226,227]. The detection of phosphorylated Zap-70, PI3K, LAT, ERK, PKC and MAPK with subsequent release of IFN-γ and TNF-α in activated Vδ2 T cells implies similar pathways to αβ TCR signaling, but does not necessarily verify the exact mechanism [228,229,230].

The intracellular signaling pathways of CD3ζ upon CAR activation in γδ T cells remain unclear, and are based on the activation pattern of Vδ2 TCR. Indirect evidence is given with the increased ERK and MAPKAPK-2 phosphorylation levels in unstimulated anti-CD33-CD28-CD3ζ CAR-Vδ2 T cells compared to unmodified ones, suggesting tonic signaling from the CAR [231].

Lack of information leads to a suboptimal understanding of intracellular CAR processes within γδ T cells, and most likely to the suboptimal persistence and killing performance of these cells when simply reusing an αβ-based CAR construct. As γδ T cells differ significantly in function, receptor expression and intracellular signaling from αβ T cells, the optimal CAR construct has yet to be engineered that is suited to the needs of γδ T cells, similarly to what has been achieved for CAR-M and CAR-NK cells.

### 5.2. Kinetics

As γδ T cells recognize antigens independently of MHC presentation, they are considered to have a low alloreactive potential and to be safe to use in an allogenic setting. Most studies utilize γδ T cells derived from peripheral blood, but the use of iPSC-derived γδ T cells as an off-the-shelf product is increasing, particularly in the context of CAR cell therapy [232,233,234]. Cord blood-derived γδ T cells, which primarily produce Vδ2 T cells, can also be successfully generated, but have not yet been used in a CAR setting [235].

In vitro expansion increases up to 72-fold within 10 days with zoledronate addition, and even up to 7042-fold when co-incubated with feeder cells over the same period [236].

In an in vivo study, anti-CD19 CAR-Vδ2 T cells were detectable in the bone marrow and spleen in NSG mice with NALM6 tumors three days after injection [237]. For anti-GPC3 CAR-Vδ1 T cells, the additional expression of IL-15 (anti-GPC3/IL15 CAR-Vδ1 T cells) boosted cell numbers in vitro after seven days compared to those without IL-15 expression [238]. In vivo, these anti-GPC3/IL15 CAR-Vδ1 T cells were primarily found in the tumor tissue by day 7, and showed limited presence in the blood, bone marrow, spleen or lungs, indicating better tumor-specific targeting compared to anti-GPC3 CAR-αβ T cells, which could be detected in all the mentioned compartments most likely due to xenoantigen reactivity and may therefore have a higher potential for causing GvHD [238]. Meanwhile, Vδ2 T cells modified with an NKG2D-CD3ζ CAR were observable in the blood of NSG mice bearing SKOV-3 tumors on day 7 [239].

Preliminary releases of clinical trial data have reported ADI-001 (anti-CD20 CAR-Vδ1 T cells) persistence in the blood for 28 days in patients with relapsed/refractory non-Hodgkin lymphoma (Figure 2). Persistence and cell proliferation were independent of HLA-matching of the product. Complete remission, or partial remission, seemed to correlate with higher peak copy numbers of ADI-001 compared to patients with stable or progressive disease [68].

Unmodified adoptively transferred Vδ2 T cells were detectable in the blood of colorectal cancer patients even 12 weeks after cell injection [240]. CAR-γδ T cells appear to have a significant yet limited duration of persistence.

### 5.3. Functions

γδ T cells possess innate tumor-killing capacities, and their cytotoxic function is enhanced by CAR transduction (Table 1) [216,217,237,241,242]. This was demonstrated by the anti-CD19-CD28-CD3ζ CAR in Vδ2 T cells, which exhibited increased cytotoxicity compared to UT γδ T cells [237]. Innate cytotoxicity was also observed in anti-CD19 CAR-γδ T cells when co-cultured with CD19-negative NALM6 cells, showing cell cytotoxicity levels comparable to those in UT γδ T cells [237]. Both CAR γδ T cells and UT γδ T cells demonstrated enhanced activity against CD19-negative NALM6 cells compared to αβ T cells transduced with the same second-generation anti-CD19 CAR [237]. Both findings suggest that γδ T cells may possess an innate killing capacity against tumor cells that is superior to αβ T cells. This killing capacity is also preserved even after CAR transduction [237]. When target antigen is expressed, CAR γδ T cells showed comparable toxicity to their αβ counterparts [237,241]. Even after dividing CAR γδ T cells into their subsets, Vδ1 T cells and Vδ2 T cells, the killing capacities of these CAR cells were found to be equivalent to each other and to conventional CAR T cells in vitro [237,241]. An in vivo comparison of CAR-αβ T cells and CAR-Vδ2 T cells, both equipped with anti-CD19-CD28-CD3ζ CARs, in NSG mice engrafted with NALM-6 leukemia demonstrated significant reductions in leukemic burden. After 14 days, residual leukemia cells in the bone marrow were reduced to 0.1% and 5%, respectively, compared to approx. 60% in untreated mice [237]. Depending on the tumor cell line and CAR construct, IFN-γ secretion can vary between cell types and constructs, with CAR-γδ T cells often secreting less than CAR-αβ T cells [241]. Further comparisons revealed that CAR-αβ T cells present a central memory (T_CM_) and effector memory (T_EM_) phenotype, whereas CAR-Vδ1 T cells maintain a naïve phenotype even during prolonged cultivation, and CAR-Vδ2 T cells are the most differentiated and mainly present a T_EM_ phenotype. CAR-Vδ1 T cells seemed to express the fewest exhaustion markers [241].

#### 5.3.1. Soluble Effector Molecules

Most effector molecules are also expressed by NK cells and are briefly described here, with detailed information available in the NK section.

γδ T cells can degranulate and release perforin, GzmB and granulysin as effector molecules to induce apoptosis in tumor cells [243,244,245]. The release mechanism is not described for γδ T cells and can only be assumed to be similar to those of αβ T or NK cells. The ability to secrete cytolytic effectors is preserved in CAR γδ T cells and was verified by CD107a and GzmB measurements after activation [217,219].

Cytokine and chemokine release slightly differ depending on whether they are induced by CAR or TCR activation [242]. TCR activation mimicked by phorbol 12-myristate 13-acetate and ionomycin induces a more diverse cytokine and chemokine profile compared to CAR, with CD28-CD3ζ domains [242]. Increased cytokine measurements were observed for IL-2, IL-4, IL-6, IL-7, IL-12, IL-15, IL-1β, IL-9, IL-10, IL-17, GM-CSF, IFN-γ and TNF-α, and for chemokines CCL2, CCL3, CCL4, CCL5, CXCL8 and CXCL10 (Figure 6) [242]. In contrast, CAR activation led to increased cytokine levels of IL-1β, IL-4, TNF-α, IL-10, IFN-γ, CCL5 and CXCL10 [242]. Both data were generated after co-cultivation with feeder cells (a clone derived from K562) and should be taken into account when interpreting the result. IL-6, IL-12, IL-15 and IL-1β most likely are detected in trace amounts due to the myeloid feeder cells. CAR activation resulted in a less diverse expression of pro-inflammatory cytokines and less IFN-γ production compared to γδ TCR activation as well as CAR αβ T cells, potentially resulting in a lower risk of CRS [237,242,246].

#### 5.3.2. Membrane-bound effectors.

γδ TCRs can broadly be divided into two different groups: Vδ2 TCRs and non-Vδ2 TCRs. Each TCR group recognizes different ligands, but both recognize ligands independently from MHC.

Vδ2 TCR can recognize conformational changes in the protein complex BTN2A1–BTN3A1 on tumor cell surfaces. BTN2A1–BTN3A1 has an intracellular sensing site, which can bind to metabolites of the isoprenoid synthesis pathway known as phosphoantigens (PAg) [247]. In tumor cells, the isoprenoid pathway is dysregulated and leads to an accumulation of isopentenyl pyrophosphate (IPP), one of the PAgs, which is recognized by BTN2A1–BTN3A1 [247,248]. Increased levels of IPP can also be achieved artificially by bisphosphonates (e.g., zoledronate), which inhibit the IPP-metabolizing enzyme farnesyl-diphosphate-synthase (FPPS) and cause metabolite accumulation [218,248]. This feature is used for Vδ2 T cell cultivation and cancer therapy. Adding zoledronate to adoptive γδ T cell infusion has increased tumor control in various studies [218].

Non-Vδ2 TCRs, especially Vδ1 TCRs, recognize various ligands (Figure 6). Vδ1 TCR can bind to MHC-like proteins, such as CD1, MR1, EPCR, MICA or MICB, which present non-peptide antigens like lipids or small metabolites. Annexin A2 and ephrin type-A receptor 2 (EphA2) are also described as Vδ1 TCR ligands. EphA2 is upregulated by metabolic changes, and can be found in cervical or colon cancer cells [247]. There is no evidence that CD1 or MR1 recognition leads to direct tumor cell killing, but as they can also present stress markers of transformed cells, they can be considered in combination with CCR or NSCAR as an “AND-gate”.

Apart from TCR, various other kinds of receptors are expressed on γδ T cells such as NK cell receptors, CD16 and TLRs (Table 1).

CD16 recognizes the Fc part of IgG antibodies and induces ADCC (Figure 6) [249]. Most of the CAR γδ studies did not explore CD16. But there are two studies: one showed that PBMC-derived CAR γδ T cells were not positive for CD16, and the other showed that iPSC-derived γδ T cells had low CD16 expression [234,242]. Further investigations are needed to determine the role of CD16 in CAR-γδ T cells.

γδ T cells express various NK-cell receptors, of which NKG2D, NKp30, NKp46 and DNAM-1 were found on CAR γδ T cells [238]. NKG2D recognizes MHC-like proteins such as MHC I chain-related molecules A and B (MICA/MICB) and UL16-binding protein (ULBP1), which are stress-induced markers on infected or transformed cells [247]. NKG2D signaling in γδ T cells is poorly understood, and its activation is most likely transmitted via the DAP10 pathway, leading to degranulation but to hardly any IFN-γ secretion [250,251]. NKp30 and NKp46 are natural cytotoxicity receptors (NCR) and are mostly detected on Vδ1 T cells (Figure 6) [247]. NKp30 recognizes B7-H6 expressed on several tumors [252]. Both receptors most likely associate with CD3ζ and lead to Ca^2+^ influx with the secretion of cytotoxic mediators and the cytokines TNF-α and IFN-γ [247,252,253]. DNAM-1 recognizes Nectin-2 or Nectin-like protein 5, which are upregulated during cellular stress and are expressed in hematological and solid tumor cells [218,254]. The intracellular signaling of DNAM-1 has not been studied in γδ T cells, but if it relates to what has been observed in NK cells, DNAM-1 activation leads to ERK and Akt activation, and subsequently to degranulation and cytotoxic effects [254].

In addition, γδ T cells express TLRs, which enable the recognition of PAMPs and respond with proinflammatory cytokines via IRF 3,5,7 and NF-κB [243]. Simultaneous activation through TCR and TLRs can enhance the effector function of γδ T cells [243].

Additionally, γδ T cells also express FAS and TRAIL as apoptotic inducers (for detailed mechanisms see NK cell section) [218]. Their expression on CAR γδ T cells has not yet been confirmed and can only be presumed.

#### 5.3.3. Antigen-Presenting Function

After IPP activation, Vδ2 T cells upregulate APC phenotypic markers, such as CD80, CD86, CD40 and HLA-DR, comparably to LPS-stimulated DCs [222,255]. Vδ2 T cells that were exposed to tetanus toxoid (TT) and mycobacterium tuberculosis–purified protein derivative (PPD) were able to induce the proliferation of TT- and PPD-specific CD4+ T cells, indicating processing within γδ T-cells and the subsequent antigen presentation of those peptides (Figure 6) [255]. Vδ2 T cells were at least as efficient as DCs concerning the proliferation induction of CD4+ T cells and CD8+ T cells, indicating an ability to cross-present [255]. The cytotoxic T lymphocytes generated upon DC or Vδ2 T cell interaction showed comparable IFN-γ and perforin secretion [255].

The HLA-DR+ and CD86+ phenotypes were observed in CAR Vδ2 T cells, and their capacities for antigen presentation were confirmed by exposure to both MART1 short- and long-peptides. The long peptides require endocytosis and processing before they can be presented to MART1-specific αβ T cells. Both types of peptides induced αβ T cell proliferation, demonstrating the CAR Vδ2 T cells’ ability to endocytose, process, and present antigens, thereby activating αβ T cells [241].

## 6. Dendritic Cells

DCs are part of the innate immunity and belong to the phagocytic system. DCs differentiate from hematopoietic stem cells into immature DCs, which possess a high endocytic capacity and low ability to activate T cells [256]. Phagocytosis, the activation of receptors and cytokine stimulation prompt DCs to mature and replace their endocytic activity with increased antigen presentation and interaction with the adaptive immune system [32,256]. Tissue-resident DCs increase CCR7 expression with maturation to migrate to secondary lymphoid organs and interact with the adaptive immune system [257]. Subsets of DCs include plasmacytoid DCs (pDCs), conventional DCs (cDCs) and inflammatory or monocyte-derived DCs (moDCs) [258,259]. The lymphoid-derived pDCs recognize viral antigens and respond with a type I interferon profile [260]. cDCs are myeloid-derived, recognize viral as well as bacterial antigens, specialize in antigen presentation and produce inflammatory cytokines and mediators such as IL-12 and TNF-α [256,260]. moDCs differentiate from circulatory monocytes under inflammatory conditions, and produce TNF-α and NO [260]. The main functions of DCs are immunosurveillance through phagocytosis, immune modulation via cytokines and chemokines, and the initialization of the adaptive immune response by antigen presentation to CD4+ T cells on MHC II, and, importantly, on MHC I to directly prime CD8+ T cells [256,258,260].

With their mobility within tissue and the above-mentioned core functions, DCs came into the spotlight of adoptive cellular therapy in cancer treatment. For decades, DCs have been used as live cell vaccines for cancer treatment due to their capacity for antigen presentation and T cell priming [261,262]. DCs are loaded with tumor-associated antigens (TAA) either directly in vivo or ex vivo, with subsequent reinfusion into the patient. TAA are processed by DCs and are presented to the adaptive immune system, inducing anti-tumor immunity [263,264,265]. The downsides of this approach include limited proliferation and loss of cells ex vivo, undirected peptide provision and unprecise antigen uptake in vivo. CAR-DCs have evolved from vaccination therapy to redirect DC to the tumor sites and induce targeted in vivo antigen uptake.

Little is known about CAR-DC, but there have been extensive preclinical and clinical live cell vaccination trials that showcase the safety and low adverse events of adoptively transferred dendritic cells [266].

### 6.1. Activation by the CAR

Vectors for CAR-DC studies are less developed due to the complexity of delivering genes into DCs. This, combined with technical challenges in isolating and manipulating DCs, makes CAR-DC studies rare and data about CAR activation scarce. One study reports that CD34+ progenitor cells and T cells were equipped with three different constructs: anti-CD33-CD137-CD3ζ CAR, CD137-CD3ζ CAR, and, as a control, eGFP. CD34+ progenitors were then differentiated into DCs using Flt3L, GM-CSF, IL-4 and AML lysates. CD34+ cells, which were transduced with a CD137 domain, showed a higher percentage of differentiation towards the CD141+CLEC9+ phenotype, which corresponds to the cDC subtype compared to the GFP control (33% vs 1.5%) [267,268]. The co-cultivation of CAR-T cells with CAR-DCs to target Kasumi-1 tumor cells demonstrated increased cytotoxicity (78.4% CAR-DC+CAR-T vs. 39.9% CAR-T vs. 17.6% eGFP T cells) and production of IFN-γ, TNF-α and IL-12 compared to CAR-T alone or the eGFP-control. CAR-T cells, together with CAR-DCs, increased the survival of NSG mice bearing AML compared to the CAR-T cells alone [267,268]. The setup complicates the assessment of the individual cell type’s contribution to the IFN-γ and TNF-α pool, but these findings suggest the activation of signaling pathways, which involve NFκB or JAK-STAT for cytokine secretion.

In another study, nanocomplexes for in vivo transfection were loaded with anti-ALK-CD28-CD3ζ-IFN-γ CAR plasmids and administered systemically in Neuro-2a tumor-bearing A/J mice [193]. This resulted in a 63.4% transfection rate of macrophages and 27.2% of DCs located in the tumor tissue [186], which mediated tumor control, and TNF-α and IFN-γ were detected in the tumor lysate [193]. However, a mixed CAR effector population complicates the assessment of the individual effector’s contribution to the mentioned effects.

### 6.2. Kinetics

Due to the nature of antigen presentation to T cells, HLA-matching is the prerequisite for DC therapy, where only autologous or HLA-matched allogeneic cells can be used. DCs can be isolated directly from the blood or differentiated from CD34+ progenitor cells or CD14+ monocytes. DC vaccination studies have used all three sources.

Little is known about the kinetics of CAR-DCs, but the data suggest that the persistence of DCs is rather short and lasts up to 21 days in vivo, as a complete turnover of DCs in several compartments was observed in a labeling study on C57BL/6J WEHI mice. There, cells of most DC subsets were only detectable up to 9–12 days [269]. A study on adoptive DC transfer in humans documented the detection of HLA-matched allogeneic peripheral blood-isolated DCs pulsed with HIV antigens up to a week after infusion (Figure 2) [69]. A study of in vivo transfected A/J mice with anti-ALK-CD28-CD3ζ-IFN-γ CAR has not reported on the persistence of CAR-DCs, but has documented detectable transfected DCs in the tumor tissue 48 h after nanocomplex injection [193].

### 6.3. Functions

As a bridge between the innate and adaptive immune system, the main function of dendritic cells is the orchestration of immunity, and therefore they can be an essential effector in the TME. DCs can enhance immune cell invasion and trafficking, and convert an immune-suppressive TME into an immunogenic zone. After antigen uptake, DCs present antigens to T cells, initiating and activating the adaptive immune system. The secretion of cytokines can change the phenotype of macrophages and enforce cytolytic functions of T cells and NK cells in the TME. Chemokine secretion recruits further immune cells such as macrophages, T cells, NK cells and granulocytes. With these functions, DCs can altogether enhance tumor immunity by modulating the TME [270].

#### 6.3.1. Endocytosis, Antigen Presentation and Immune Modulation

DCs impart immunosurveillance through endocytosis, antigen-presentation and immune modulation. DCs can use several pathways for antigen uptake: macropinocytosis, receptor-mediated endocytosis and phagocytosis [271]. Antigen-presentation can occur via phagolysosomal degradation and MHC I/II loading or cross-dressing. Such an ability for endocytosis and antigen cross-presentation was employed in a study where a CAR termed extracellular vesicle (EV)-internalizing receptor (EVIR) was constructed to capture and present tumor antigens on DCs [272]. The EVIR consisted of an anti-HER2 scFv domain and a truncated (non-signaling) low-affinity nerve growth factor receptor (dLNGFR) [272]. It was compared to a control CAR without the scFv domain [272]. EVs are secreted by cancer cells and include microvesicles, which are generated by the outward budding of the membrane and exosomes and contain cell surface, cytosolic and nuclear proteins, RNA transcripts, micro-RNAs and DNA fragments as cargo [273]. EVs are used for intercellular communication and are used to influence metastasis, cell growth and survival, or to coordinate the TME [273]. Transduced CAR-DCs were incubated with EVs from HER2 or non-HER2-expressing and mCherry-labeled MC38 cancer cells. EVIR-DC efficiently took up HER2+ mCherry-labeled EVs compared to non-HER2-expressing EVs or control DCs without scFv [272]. The fluorescence shifted from membrane-bound to diffuse dissemination within the cell over time, indicating the binding and internalization of the EVs [272]. Here, EVIR-DCs showed micropinocytosis as a means of EVIR-bound EV internalization [272].

Cross-presentation by DCs was verified by pre-incubating with EVs from HER2- and ovalbumin (OVA)-expressing MC38 cells labeled EV-HER2/OVA and EV-OVA. These DCs were then co-cultured with CD8+ OT-I-T cells that recognize the OVA-derived SIINFEKL peptide on MHC I, leading to increased OT-I cell proliferation when pre-incubated with EV-HER2/OVA compared to pre-incubation with EV-OVA. This setup, when also applied in a syngeneic in vivo model, demonstrated enhanced tumor control and the proliferation of tumor-specific CD8+ T cells, illustrating a process known as cross-dressing in antigen presentation [272].

#### 6.3.2. Soluble Effector Molecules

Except for the study on the anti-CD33-CD137-CD3ζ CAR that has provided data on IFN-γ, TNF-α and IL-12 secretion, no further information on cytokine or chemokine secretion is accessible [267,268]. However, even these data do not conclusively pinpoint the exact cellular source of IFN-γ and TNF-α secretion, as the cytokines were measured in the supernatant of a co-cultivation with T cells [267,268]. The next section therefore only summarizes the physiological secretions of DCs (Table 1).

##### Cytokines, Chemokines, Exosomes

Cytokine expression differs between the DC subsets, but the spontaneously expressed cytokines include IL-1α, IL-1β, IL-6, IL-7, IL-12 (p35 and p40), IL-15, IL-18, TNF-α, TGF-β, M-CSF, and GM-CSF, whereas IL-2, IL-3, IL-4, IL-5, IL-9, and IFN-γ transcripts are not typically detected [274]. DCs, especially cDC1, are instrumental in relaying expansion signals and immunomodulatory effects to cytotoxic lymphocytes (CTLs) and NK cells via IL-12, IL-15 and IL-18 [275]. Cytokines like Flt3-L are crucial for DC precursor attraction and local expansion in TME [259]. Leveraging this, DCs have been engineered to co-express IL-12 with Flt3-L or IL-18, enhancing their ability to recruit and activate antitumor effectors and endogenous cDC1s. This modification bolstered antigen-dependent responses and amplified antigen-independent pathways, improving the anti-tumoral immune landscape [276,277].

As for chemokines, moDCs and CD34+-derived DCs can secrete CCL9, CCL18, CCL22 and CCL25 [259,278]. Mature DC induced by CD40, or inflammatory agonists, can additionally secrete CXCL8, CCL2 and CCL19 [278]. Most of the chemokines released by DCs recruit B and T cells, macrophages, monocytes, neutrophils and other DCs [278]. CXCL9, CXCL10 and CXCL11, for example, attract NKs and activated T cells into tumor tissue [259]. These NKs and innate lymphoid cells in turn reinforce the NK–DC axis by secreting CCL5 and XCL1, which recruit and further activate XCR1+ cDC1s [259].

DCs can also release exosomes (Dexs), which transport MHC/peptide complexes, co-stimulatory molecules (CD80, CD86), integrins and ICAMs [279]. These Dexs can interact with immune cells, inducing antigen-specific responses in already-activated T cells but not in naïve T cells [279]. Additionally, Dexs facilitate the cross-dressing of bystander DCs, enhancing their antigen presentation capacity [279].

#### 6.3.3. Membrane-Bound Effectors

Staining for surface markers has not been included in the CAR-DC studies currently available; therefore, the next section only contains a summary of the physiological receptors of DCs.

DCs express phagocytotic receptors, which can induce receptor-mediated antigen uptake (Figure 6). They include C-type lectin receptors (DC-SIGN, CD205, CD206, CLEC9A), Fcγ receptor types (CD64, CD32) and scavenger receptors (CD36) (Table 1) [271]. DC-SIGN is internalized upon antigen binding. It possesses a YXXΦ domain (Y is tyrosine, X is any amino acid, and Φ is an amino acid with bulky hydrophobic side chain receptors) and an acidic triad (EEE), which serve as internalization and sorting signals for the distribution to endosomes and lysosomes [280,281]. CD206 and CD205 are constitutively endocytosed through tyrosine-based motifs FENTLY and FSSVRY, respectively [280,282]. After internalization, the receptor–antigen complex is redirected to endosomes or lysosomes. The receptor is recycled, and the antigen is processed and loaded on MHC I/MHC II complexes for antigen presentation [282]. Fcγ receptor-mediated internalization is ITAM- and Rac1-dependent [283,284].

DCs express several PRRs, including the already-mentioned C-type lectin receptors (CLRs), TLRs, nucleotide-binding domain- and leucine-rich repeat-containing family (NLRs), and retinoic acid-inducible gene I (RIG-I)-like receptors (RLRs) [285].

Human pDC expresses TLR7 and TLR9, while the other DC subsets express TLR1, TLR2, TLR3, TLR5, TLR6, and TLR8 [286]. Monocyte-derived DCs progressively lose TLR1, TLR2, TLR4, and TLR5 and acquire TLR3 expression during differentiation with GM-CSF and IL-4 [287]. TLR signaling induces proinflammatory cytokine production, such as TNF-α, IL-12 or type I IFN, depending on the TLR type, and it enhances antigen presentation to naïve T cells (see macrophage section for detailed information on TLR signaling) [286].

CLR signaling can be split into two types: activating and inhibiting pathways. The activating pathway is regulated by Dectin-1 and DC-SIGN. For activating pathways, Dectin-1 utilizes the YxxL motif to recruit SYK kinase. This in turn assembles a complex with CARD9, BCL-10 and MALT1, leading to the activation of the canonical NF-κB pathway [279]. Additionally, SYK can initiate an alternative pathway by recruiting NIK, which activates the non-canonical NF-κB pathway. Conversely, DC-SIGN activates its pathway by engaging Ras, which subsequently activates Src kinases and p21-activated kinases. These kinases phosphorylate Raf1, leading to distinct NF-κB activation, without triggering ERK activation [279]. Both activating pathways can enhance TLR signaling.

The inhibitory pathways involve the receptors DCIR and CLEC4C, which operate through ITIM or the ITAM-SYK-CARD9/BLNK/PLCγ2 cascade. This induces tonic calcium signaling, which inhibits MYD88 recruitment, consequently downregulating TLR signaling [279].

The triggering of RLRs induces CARD conformation change and the activation of IPS-2 signalosome activation, which translocates IRF3 and IRF7 into the nucleus, leading to an IFN type I response [288].

CD40 activation leads to TRAF recruitment into the cytoplasmic tail. DCs can recruit TRAF2, 3, and 5, but predominantly use TRAF6 [289]. TRAF6 is also used in TLR signaling and induces MAPK and JNK activation with subsequent IL-12 and IL-6 production, as well as NF-κB translocation [289,290]. CD40 can also lead to an IRF3/IRF7-independent IFN type I response; however, this pathway is still under investigation [289].

## 7. Conclusions

Because of the treatment success of CAR-T cells in hematological malignancies, as well as their frequent application and long history of use, the advantages but also the limitations of CAR-T cells are very well outlined. The weaknesses concern production procedures (facilities, logistics, time-point coordination, expansion), CAR engineering (exhaustion, persistence, on-target off-tumor effect, toxicities) and cell properties (viability, HLA limitation, functions, GvHD, HvG rejection, susceptibility to immune suppression, scant TME and tumor tissue infiltration). In addition, the various malignant entities with their variable properties (dense, immune-suppressive, metabolic hostile TME; antigen-escape mechanisms; HLA downregulation; pathway dysregulation) cause challenges to adoptive cell therapy.

These adversities have led to an expansion of the field to enhanced engineering (transgenic proteins, switch receptors, “AND” gate receptors, split CARs), cell types (adaptive and innate immune cells) and cell products (cell lines, stem cells, cord blood, induced pluripotent stem cells, off-the-shelf usage), but space remains for optimization.

Up until now, the CD3ζ CAR, which has been optimized for T cells, has been used for CAR transduction in almost every cell type. But given the different functions and distinct intracellular signaling pathways of every cell type, the same signaling domain does not necessarily lead to an optimal performance in every context. The search for and the creation of ideal immunotherapeutic adoptive cell products are still ongoing. Adoptive cell therapy has met its limitations, but has not hit its ceiling yet. Innovations, particularly those tailored to the specific needs of each immune cell type, will be needed to overcome or exploit their unique quirks and properties.

## Figures and Tables

**Figure 1 cancers-16-02608-f001:**
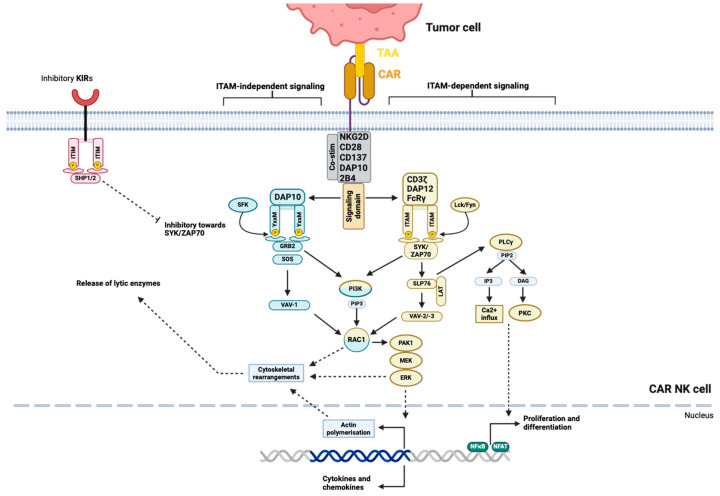
Intracellular signaling pathways in NK cells after CAR activation with different signaling domains. CAR stimulation can lead to two main pathways depending on the signaling domain. CD3ζ, DAP12, and FcRγ signal through ITAM. DAP10 signals through YxxM and is ITAM-independent. Pathways are color-coded. Created with BioRender.com.

**Figure 2 cancers-16-02608-f002:**
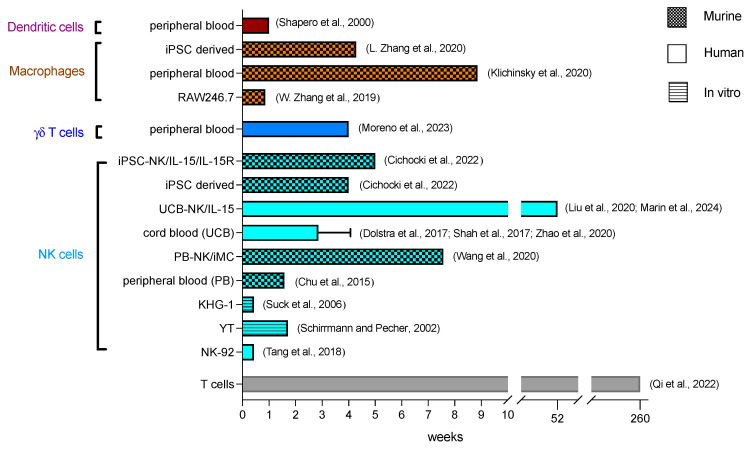
The persistence of different cell effectors in vitro and in vivo. Depicted in weeks. In vivo data contain murine or human data. Data include cell effectors from cell lines, primary cells and stem cells. NK cells in light blue, γδ T cells in dark blue, macrophages in orange, and dendritic cells in purple. In vitro data shown with the lined pattern, murine data shown with the chequered pattern. Human data shown without a pattern [24,54,55,56,57,58,59,60,61,62,63,64,65,66,67,68,69]. Persistence data present data from confirmed cell detection at the specified time point. Persistence beyond this time point is possible but speculative.

**Figure 3 cancers-16-02608-f003:**
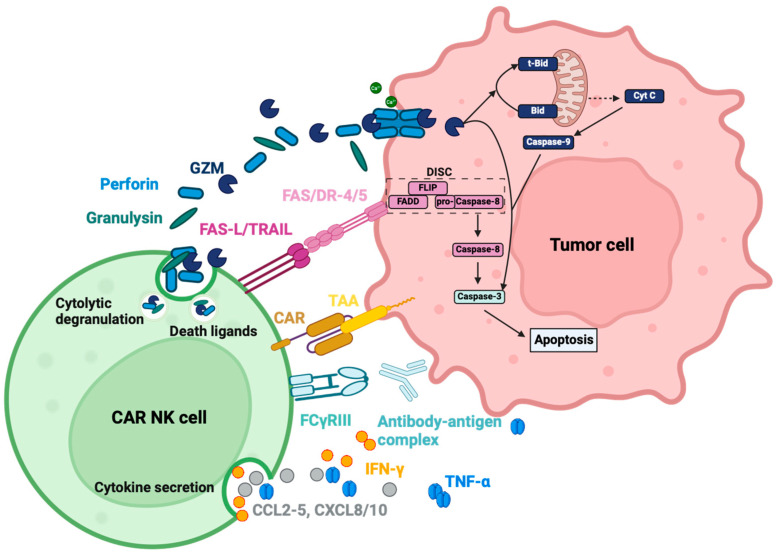
CAR-NK cell effector function against tumor cells. Depiction of the immunological synapse and soluble and membrane-bound effector molecules. CAR activation induces cell activation with the release of cytolytic granules and cytokines/chemokines. Additional apoptosis induction with FASL/TRAIL expression and ADCC via FcγRIII receptors. Created with BioRender.com.

**Figure 4 cancers-16-02608-f004:**
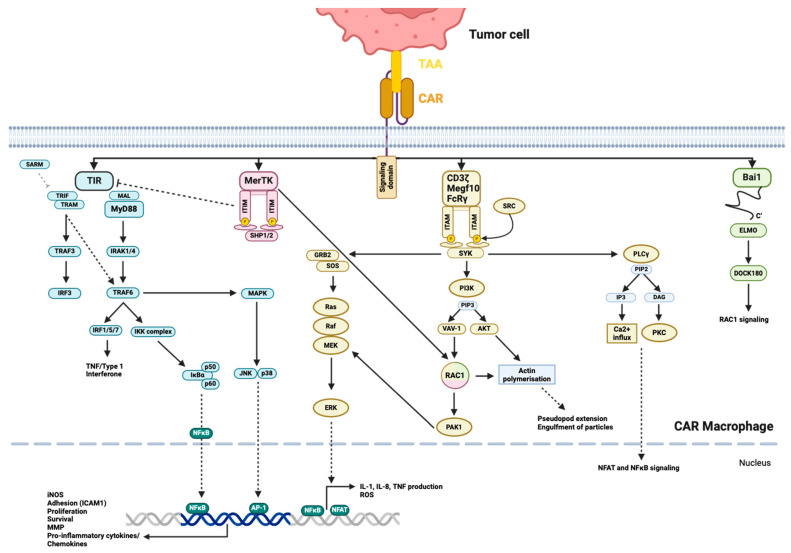
Intracellular signaling pathways in macrophages after CAR activation with different signaling domains. CAR stimulation can lead to two main pathways depending on the signaling domain. CD3ζ, Megf10, FcRγ, Bai and MerTK pathways converge on Rac1 and induce actin polymerization and phagocytosis. TIR mainly induces iNOS, chemokines, cytokines, and survival and proliferation signals. Pathways are color-coded. Created with BioRender.com.

**Figure 5 cancers-16-02608-f005:**
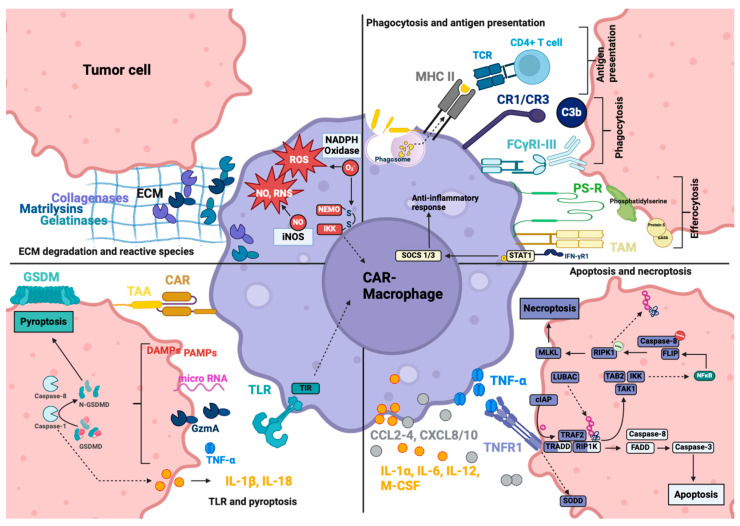
CAR-M cell effector functions in relation to tumor cells. The depiction of soluble and membrane-bound effector molecules and their functions. The main functions of CAR-M are phagocytosis induced by phagocytosis/efferocytosis receptors, antigen presentation via MHC II, ECM degradation by MMP, ROS production, apoptosis, necroptosis, pyroptosis, cytokine and chemokine release. Created with BioRender.com.

**Figure 6 cancers-16-02608-f006:**
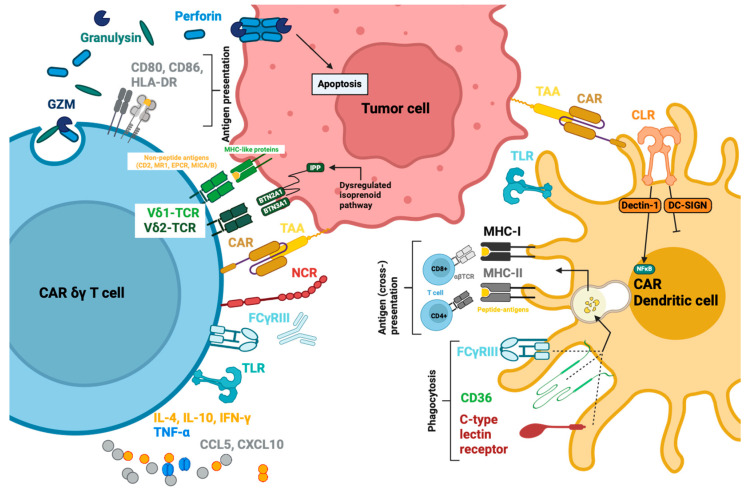
Effector functions of CAR γδ T and dendritic cells in relation to tumor cells. Depiction of soluble and membrane-bound effector molecules. Main functions of CAR γδ T cells: killing capacity with cytolytic granules after CAR activation, innate killing capacity with FCγRIII via ADCC and NK cell receptors, antigen presentation and cytokine and chemokine release. Main functions of dendritic cells: endocytosis, antigen (cross-) presentation, cytokine and chemokine release. Intracellular signalling pathways are not depicted due to insufficient published data and information. Created with BioRender.com.

**Table 1 cancers-16-02608-t001:** Summary of CAR cells’ mode of action.

Cell Type	Effector Molecules		Signaling Domain/MOTIF		Antigen Recognition/ Autologous/Allogenic	Cell Source
	Soluble	Membrane-Bound				
CAR-T	Perforin-1	FasL	CD3ζ	ITAM	MHC-dependent	PB
	GzmA/B	TRAIL			autologous	
	Granulysin	TCR			HLA-matched	
	IFN-γ					
	TNF-α					
CAR-NK	Perforin-1	FASL	CD3ζ	ITAM	MHC-independent	PB
	GzmA/B/H/M	TRAIL	FcRγ	ITAM	autologous	UCB
	Granulysin	CD16	DAP12	ITAM	allogenic	iPSC
	TNF-α	NKG2D	DAP10	YXXM	HLA-(mis)matched	Human embryonic SC
	IFN-γ	NCR				BM
	CXCL8/10					YTS
						KHYG-1
	CCL2/3/5					NK92

CAR-M	TNF-α	FcγR	CD3ζ	ITAM	MHC-independent	PB
	IL-1α/1β/18	CD36 CD14	FcRγ	ITAM	autologous	UCB
					allogenic	iPSC
	IL-6/12/23	TLR-3/4/7/8/11	Megf10	ITAM	HLA-(mis)matched	BM
						THP-1
	CCL2/3/4/8	NLR	MerTK			U937
	CXCL8/10	αvβ3-Integrin	Bai1			J774A.1
			TIR			RAW264.7
	MMP-3/9/10/11/12/13/14	TAM	CD147			
		CR1/3	P13K			
		SR-A				
		MARCO				
	ROS/RNS/NO					
Additional functions	Phagocytosis					
	Antigen presentation (MHC II)					
	Apoptosis, pyroptosis, necroptosis					
	TME-remodeling					
CAR-γδ T	Perforin-1	Vδ1 TCR	CD3ζ	ITAM	MHC-independent	PB
	GzmB	Vδ2 TCR	DAP10		autologous	iPSC
	Granulysin	NKG2D	Truncated/NS		allogenic	
	IFN-γ	NKp30			HLA-(mis)matched	
	TNF-α	NKp44				
	IL-4	DNAM-1				
	IL-10	TLR				
	CCL5	CD16				
	CXCL10	FasL				
		TRAIL				
Additional functions	Antigen presentation MHC I/MHC II					
CAR-DC	TNF-α	DC-SIGN	CD3ζ	ITAM	MHC-dependent	PB
	GM-CSF	CD205	Truncated/NS		autologous HLA-matched	
	M-CSF	CD206				
	TGF-β	Dectin-1				
	IL-1α/1β/18	CLEC9A				
	IL-6/7/12/15	RLR				
	CCL-2/9/18/19/22/25	CD40				
		TLR-1/2/3/4/5/6/8				
	CXCL8					
	Dex					
Additional function	Macropinocytosis, receptor-mediated endocytosis, phagocytosis					
	Antigen presentation (MHC I/MHC II)					
	Cross-dressing					
	Immune modulation					

Table also includes physiological effector molecules and is not restricted to effectors proven on CAR-Cells. Gzm = Granzyme; PB = peripheral blood; UCB = umbilical cord blood; iPSC = induced pluripotent stem cells; SC = stem cells; BM = bone marrow; Dex = dendritic cell exosome; MMP = matrix metalloprotease; ROS = reactive oxygen species; RNS = reactive nitrogen species; NS = non-signaling; ITAM = immunoreceptor tyrosine-based activation motifs. Table also includes physiological effectors not.

## Data Availability

No original data have been generated in this review.

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
