# Peer review of "The Spectrum of CAR Cellular Effectors: Modes of Action in Anti-Tumor Immunity"

_cancers, 2024, doi:10.3390/cancers16142608_

Round 1

Reviewer 1 Report

Comments and Suggestions for Authors

In this manuscript, the authors compared the different cell types and specific modes of action upon activation of the CAR in these cell types, even though, which could help clinicians and researchers understand of their peculiarities and commonalities to the currently broadly used T cells, even though, several concerns should be further addressed about this manuscript.

Major revisions:

1.    In lines 70-75 of the manuscript, the author described "In 1986 the first tumor-infiltrating lymphocytes (TILs) were isolated out of melanoma lesions," but in fact, Professor Rosenberg had successfully isolated TILs and confirmed the function of TILs by vitro experiments in 1979. Please correct the description of TILs.

2.    In lines 83-86 of the manuscript, the author described " This was, therefore, the next hurdle to overcome and resulted in a fusion protein consisting of an antigen recognition domain derived from antibodies and the intracellular TCR domain creating the first chimeric antigen receptor (CAR) and CAR-T cell, which was then in 1993 called “T-body””. But in fact, the research “Expression of immunoglobulin-T-cell receptor chimeric molecules as functional receptors with antibody-type specificity.” had mentioned the “Chimeric T cell receptor molecular” , please correct the description.

3.    In lines 83-86 of the manuscript, the author described “the intracellular signaling domain mostly consists of the CD3ξ domain of the TCR with the immunoreceptor tyrosine-based activation motif (ITAM) as signal transducer”, but CD3ξ is not a domain of TCR, CD3ξ and TCR form a complex which TCR signaling dependent on the ITAM of CD3ξ, please correct the description.

4.    Review enables researchers and clinicians to learn about this field more quickly, but little information is provided such as only the peak of expansion and organ distribution were mentioned in the "2. T-lymphocytes" CAR T cell dynamics. Up to now, there are many studies have established efficacy and characteristics of CAR T products, please make a point about indicators related to clinical efficacy, such as CAR T phenotypes associated with sustained anti-tumor treatment and characteristics of premanufacture CD8 + T cells from patients so that to play a guiding role.

5.    Please add dates of clinical trials in the section "3. Natural killer cells" to help readers quickly understand the application of CAR-NK.

6.    In the section "4. Macrophages and 6. Dendritic cells", the author detailed the function and secretion of CAR-M/DC, but different from T cells and NK cells, macrophages and DC cells enhance anti-tumor immunity by modifying the tumor microenvironment. Please review the relationship between secretion and anti-tumor immunity which could help readers quickly understand functions of CAR-M/CAR-DC.

Minor revisions:

1.    There were no descriptions about Figures there were two “Figure.1” in this manuscript, please check about it.

2.    There were many "Error! Reference source not found.” in the manuscript, to ensure the accuracy of information please refer to the original study, rather than directly quoting statements about the study from other reviews.

Comments on the Quality of English Language

Author Response

Reviewer 1

Comments 1: In lines 70-75 of the manuscript, the author described "In 1986 the first tumor-infiltrating lymphocytes (TILs) were isolated out of melanoma lesions," but in fact, Professor Rosenberg had successfully isolated TILs and confirmed the function of TILs by vitro experiments in 1979. Please correct the description of TILs.

Response 1: Thank you for pointing this out. Our description was very confusing. For 1986 we wanted to describe the adoptive cell therapy with TIL in mice and not the first isolated TIL. The statement has been corrected: “The successful isolation of tumour infiltrating lymphocytes (TILs) followed and in 1986 ex vivo expanded TILs were injected as a living drug into mice bearing MC-38 colon adenocarcinoma cells, which in combination with interleukin-2 (IL-2) and cyclophosphamide or total body irradiation cured the mice of metastatic lesions in the liver and lung” (lines 71-74)

Comments 2: In lines 83-86 of the manuscript, the author described " This was, therefore, the next hurdle to overcome and resulted in a fusion protein consisting of an antigen recognition domain derived from antibodies and the intracellular TCR domain creating the first chimeric antigen receptor (CAR) and CAR-T cell, which was then in 1993 called “T-body””. But in fact, the research “Expression of immunoglobulin-T-cell receptor chimeric molecules as functional receptors with antibody-type specificity.” had mentioned the “Chimeric T cell receptor molecular” , please correct the description.

Response 2: Thank you for pointing this out. The phrasing was not specific enough. We were referring to the first actual first-generation CAR fused with a CD3z domain in 1993. The phrasing has been specified to avoid confusion: “the next hurdle to overcome and resulted in a fusion protein consisting of an antigen recognition domain derived from antibodies and the intracellular CD3z signalling domain of the TCR/CD3 complex creating the first-generation chimeric antigen receptor (CAR) and CAR-T cell, which was then in 1993 called “T-body” by the Eshhar Group” (lines 86-90)

Comments 3:  In lines 83-86 of the manuscript, the author described “the intracellular signaling domain mostly consists of the CD3ξ domain of the TCR with the immunoreceptor tyrosine-based activation motif (ITAM) as signal transducer”, but CD3ξ is not a domain of TCR, CD3ξ and TCR form a complex which TCR signaling dependent on the ITAM of CD3ξ, please correct the description.

Response 3: Thank you for pointing this out. The statement was inaccurate. We have corrected the term to “TCR/CD3 complex”: “signalling domain mostly consists of the CD3z domain of the TCR/CD3 complex with the immunoreceptor tyrosine-based activation motif (ITAM)” (now in line 97)

Comments 4: Review enables researchers and clinicians to learn about this field more quickly, but little information is provided such as only the peak of expansion and organ distribution were mentioned in the "2. T-lymphocytes" CAR T cell dynamics. Up to now, there are many studies have established efficacy and characteristics of CAR T products, please make a point about indicators related to clinical efficacy, such as CAR T phenotypes associated with sustained anti-tumor treatment and characteristics of premanufacture CD8 + T cells from patients so that to play a guiding role.

Response 4: Thank you for this input. We have added information on T-cell phenotype and CAR composition and now refer to publications and reviews which will inform the interested reader in greater detail about this important topic for CAR-T cells.

“Altogether, persistence is an indicator tightly linked to clinical efficacy and can be in-fluenced by T cell phenotype and T cell fitness. T cell subsets such as central memory T cells or stem cell memory T cells were more prevalent in patients with sustained anti-tumour response [26]. Selection of these specific subsets and production of CAR-T cells with defined CD4+ and CD8+ phenotypes and compositions can therefore provide uni-form potency and superior antitumor activity [27–30]. In contrast prolonged and recur-rent exposition to high tumour burden in relapsed/refractory cancer patients lead to an exhausted phenotype of T cells and to an impaired CAR-T cell product [31].” (lines 192-199)

Comments 5: Please add dates of clinical trials in the section "3. Natural killer cells" to help readers quickly understand the application of CAR-NK.

Response 5: Thank you for this input. We have added the information as follows to clinical trials mentioned in the NK section (example): “(NCT02944162, phase I trial of CD33-CAR NK-92 cells in three patients with RR-AML)” (line 308,352,355)

Comments 6: In the section "4. Macrophages and 6. Dendritic cells", the author detailed the function and secretion of CAR-M/DC, but different from T cells and NK cells, macrophages and DC cells enhance anti-tumor immunity by modifying the tumor microenvironment. Please review the relationship between secretion and anti-tumor immunity which could help readers quickly understand functions of CAR-M/CAR-DC.

Response 6: Thank you for this input. We have added a section in the macrophage and DC section to summarise the influence of these cells on the microenvironment.

Macrophages: “Unlike CAR-T and CAR-NK cells, CAR-M can additionally influence and modulate the TME. Secreted cytokines can activate bystander immune cells and induce phenotype change of TAMs, promote antigen uptake, maturation and presentation of DCs or en-hance cytolytic functions of NK and T cells. Chemokines recruit further immune cells to the tumour site and MMPs can influence permeability of TME, facilitating immune cell invasion.” (lines 825-831)

Dendritic Cells: “As a bridge between innate and adaptive immune system, the main function of dendritic cells is the orchestration of immunity and therefore can be an essential effector in the TME. DCs can enhance immune cell invasion and trafficking and convert an immune-suppressive TME into an immunogenic zone. After antigen uptake, DCs pre-sent antigens to T cells initiating and activating adaptive immune system. The secre-tion of cytokines changes the phenotype of macrophages and enforces cytolytic func-tions of T cells and NK cells in the TME. Chemokine secretion recruits further immune cells such as macrophages, T cells, NK cells and granulocytes. With these functions DCs can altogether enhance tumour immunity by modulating the TME [273].” (lines 1422-1430)

Minor revisions:

Comments 1: There were no descriptions about Figures there were two “Figure.1” in this manuscript, please check about it

Response 1: Thank you for pointing this out. There was a problem with the referencing software and the cross-referencing function of word. We have re-inserted the figure captions, cross-references and the references to solve the problem.

Comments 2: There were many "Error! Reference source not found.” in the manuscript, to ensure the accuracy of information please refer to the original study, rather than directly quoting statements about the study from other reviews.

Response 2: Thank you for pointing this out. There was a problem with the referencing software and the cross-referencing function of word. We have re-inserted the figure captions, cross-references and the references to solve the problem. The references have been re-checked for original studies.

Reviewer 2 Report

Comments and Suggestions for Authors

Nguyen et al. present a very detailed manuscript that provides description of many aspects of the immune cells being investigated to use in CAR-immune cell therapies. As well as providing a thorough background on the immune cells themselves, it also describes the CAR-mediated therapies associated with each immune cell type. This paper could be published as a resource for the field; however, the following minor concerns should be addressed:

Comments:

11. The Simple Abstract and Abstract are very similar with a good portion having word-for-word exact phrasing.

  2. The Figures are mislabeled and do not have call-outs in the text which makes it confusing for the reader to follow along. Also related to the figure formatting:

     (a) The figure on page 2 has no figure number or legend and is confusing to where it fits into the manuscript. Figure 1 is labeling 2 different figures.

     (b) The author has bolded part of the words in the first sentence of some of the figure legends. If this is intentional, the authors should be more consistent.

  3. Some of the language is a bit vague. For example, on page 4 line 124, the authors state the CAR T cells are armed with “other transgenic proteins”. It might be helpful to include an example of what the authors are referring to.

  4. There appears to be an error with the referencing software and so the citations have not been documented correctly. This occurs throughout the manuscript.

  5. IFN-g is sometimes mis-abbreviated as INF-g.

Author Response

Reviewer 2

Comments 1: The Simple Abstract and Abstract are very similar with a good portion having word-for-word exact phrasing.

Response 1: Thank you for pointing this out. The simple summary has been re-written and has been further simplified. “Chimeric antigen receptor (CAR)-T cells have revolutionized the treatment of blood cancers. However, their effectiveness faces challenges in solid tumours. Modifications to CAR-T cells al-tered stimulatory domains or added further functions with transgenic proteins (e.g. cytokines, chemokine receptors, degrading enzymes) to match specific barriers of the tumour microenvi-ronment. But despite these enhancements, inherent limitations of CAR-T cells such as dependency on human leukocyte antigen (HLA), toxicities and high costs for preparing autologous cell prod-ucts remain. In response, alternative types of immune cells with different effects and advantages have become the focus of adoptive cell therapy research. For instance, natural killer cells have the benefit of HLA-independent killing and macrophages possess additional functions such as phag-ocytosis and antigen-presentation. As these cells come with distinct properties, clinicians and re-searchers need a thorough understanding of their effects and peculiarities. This review summa-rizes the different modes of action of these CAR-immune cells.” (lines 23-34)

Comments 2: The Figures are mislabeled and do not have call-outs in the text which makes it confusing for the reader to follow along. Also related to the figure formatting:

(a) The figure on page 2 has no figure number or legend and is confusing to where it fits into the manuscript. Figure 1 is labeling 2 different figures.

(b) The author has bolded part of the words in the first sentence of some of the figure legends. If this is intentional, the authors should be more consistent.

Response 2: Thank you for pointing this out. There seemed to be a problem with the caption function and cross-reference function of word. The pictures, captions and call-outs in the text have been re-inserted and re-added to solve the problem.

(a) Thank you for pointing this out. The figure on page 2 is the graphical abstract and will appear alongside with the text abstract in the Table of Contents and not like depicted right now in the main text. This will be applied by the journal after acceptance and will hopefully resolve any further confusion. The numbering of the figures has been corrected.

(b) Thank you for pointing this out. The figure legends have been changed into a uniform format.

Comments 3: Some of the language is a bit vague. For example, on page 4 line 124, the authors state the CAR T cells are armed with “other transgenic proteins”. It might be helpful to include an example of what the authors are referring to.

Response 3: Thank you for pointing this out. We have added the examples: “armouring CAR-T cells with transgenic proteins such as cytokines, chemokine receptors or matrix degrading enzymes” (lines 125,126)

Comments 4: There appears to be an error with the referencing software and so the citations have not been documented correctly. This occurs throughout the manuscript.

Response 4: Thank you for pointing this out. There was indeed a problem with the referencing software and the cross-referencing function of word. We have re-inserted the necessary information to solve the problem.

Comments 5: IFN-g is sometimes mis-abbreviated as INF-g.

Response 5: Thank you for pointing this out. We have changed the abbreviations thoughout the text.

Round 2

Reviewer 1 Report

Comments and Suggestions for Authors

The authors have already addressed all my concerns. suggest to be formally accepted.